# VESPA: an optimized protocol for accurate metabarcoding-based characterization of vertebrate eukaryotic endosymbiont and parasite assemblages

Leah A. Owens [1] ✉, Sagan Friant [1,2,3], Bruno Martorelli Di Genova [4,5], Laura J. Knoll [4], Monica Contreras[6], Oscar Noya-Alarcon[7], Maria G. Dominguez-Bello [8,9,10,11] & Tony L. Goldberg [1] ✉

Protocols for characterizing taxonomic assemblages by deep sequencing of short DNA barcode regions (metabarcoding) have revolutionized our understanding of microbial communities and are standardized for bacteria, archaea, and fungi. Unfortunately, comparable methods for host-associated eukaryotes have lagged due to technical challenges. Despite 54 published studies, issues remain with primer complementarity, off-target amplification, and lack of external validation. Here, we present VESPA (Vertebrate Eukaryotic endo-Symbiont and Parasite Analysis) primers and optimized metabarcoding protocol for host-associated eukaryotic community analysis. Using in silico prediction, panel PCR, engineered mock community standards, and clinical samples, we demonstrate VESPA to be more effective at resolving host-associated eukaryotic assemblages than previously published methods and to minimize off-target amplification. When applied to human and non-human primate samples, VESPA enables reconstruction of host-associated eukaryotic endosymbiont communities more accurately and at finer taxonomic resolution than microscopy. VESPA has the potential to advance basic and translational science on vertebrate eukaryotic endosymbiont communities, similar to achievements made for bacterial, archaeal, and fungal microbiomes.

Microbiomes are multikingdom assemblages of microorganisms and their entire "theater of activity" including signaling molecules and metabolites[1]. Such communities have emergent properties arising from cross-species and cross-kingdom interactions[2]. One of the most salient examples is the human gut, wherein bacterial community dynamics have direct effects on health[3] and can be manipulated to improve disease outcomes in clinical settings[4]. Evidence is mounting that assemblages of host-associated eukaryotes also form

[1]Department of Pathobiological Sciences, School of Veterinary Medicine, University of Wisconsin-Madison, Madison, WI, USA. [2]Department of Anthropology, The Pennsylvania State University, University Park, PA, USA. [3]Huck Institutes of the Life Sciences, The Pennsylvania State University, University Park, PA, USA. [4]Department of Medical Microbiology and Immunology, University of Wisconsin-Madison, Madison, WI, USA. [5]Department of Microbiology and Molecular Genetics, Larner College of Medicine, The University of Vermont, Burlington, VT, USA. [6]Center for Biophysics and Biochemistry, Venezuelan Institute of Scientific Research (IVIC), Caracas, Venezuela. [7]Centro Amazónico de Investigación y Control de Enfermedades Tropicales-CAICET, Puerto Ayacucho, Amazonas, Venezuela. [8]Department of Biochemistry and Microbiology, Rutgers University–New Brunswick, New Brunswick, NJ, USA. [9]Department of Anthropology, Rutgers University, New Brunswick, NJ, USA. [10]Institute for Food, Nutrition and Health, Rutgers University, New Brunswick, NJ, USA. [11]Canadian Institute for Advanced Research (CIFAR), Toronto, ON, Canada. ✉e-mail: leah.owens@wisc.edu; tony.goldberg@wisc.edu

communities with important consequences for host health[5], although they are far less studied compared to their bacterial, archaeal, and fungal counterparts[6]. Even terminology to describe host-associated eukaryotes is lacking. "Eukaryotic microbiome/microbiota"[7] does not include host-associated macro-organisms such as helminths, "nemabiome"[8] is limited to nematodes, and "parasites"[9] excludes commensal/beneficial organisms and includes ectoparasites. Herein we use the term "eukaryotic endosymbionts" to refer to both host-associated microscopic eukaryotes (microsporidia, protozoa, algal parasites) and macroscopic metazoans (helminths, pentastomes). In this context, we use the prefix -endo to include endoparasites and commensals, while excluding ectoparasites (mites, ticks, fleas). We exclude fungi because of their fundamentally different life histories[10] and the fact that established methods already exist for assessing the "mycobiome"[11]. However, we include microsporidia, because their life cycles are considered more similar to protozoa than to fungi[12].

Well-established methods exist to study eukaryotic endosymbiotic organisms. Microscopic observation has been an essential tool since van Leeuwenhoek first described *Giardia* in the seventeenth century[13]. Combined with subsequent advances in staining and enrichment techniques, microscopy is still a gold standard method[14], although it requires specialized training[15] and has inherent resolution limits (i.e., some species cannot be distinguished solely based on morphology, a phenomenon known as "cryptic species complexes"[16]). For example, the genus *Entamoeba* contains pathogenic *E. histolytica* and benign *E. dispar* which appear identical under the microscope[17]. More recently developed molecular assays (e.g., PCR and DNA sequencing of amplicons) have enabled finer taxonomic differentiation, including strain-level identification of species complexes[18]. Although useful, such assays usually have high DNA sequence specificity and are therefore not suitable for characterizing diverse assemblages of eukaryotic endosymbionts.

Methods for characterizing bacterial and fungal assemblages are standardized and based on massively parallel sequencing of amplified marker genes, or metagenomic barcoding (henceforth metabarcoding)[19]. For bacteria, the 16S ribosomal RNA (16S rRNA, or just 16S) locus[20] and for fungi, the internal transcribed spacer (ITS) locus[21] are proven targets for metabarcoding. By contrast, universal targets and protocols for metabarcoding of eukaryotic endosymbionts are not standardized[6]. For example, some published methods utilize PCR primer sets originally designed for free-living eukaryotic microbes[22–25], some target metazoans only[26,27], while others focus exclusively on helminths[8,28–30] or gut-associated organisms[31–33]. There is also a conspicuous absence of published comparisons to gold standard methods such as microscopy[34]. Moreover, no commercially available reagents exist for assessing the accuracy of eukaryotic endosymbiont metabarcoding-based methods. Community standards (mixtures of organisms or their genetic material in known composition and quantity) have been important for standardizing microbiome protocols and are commercially available[35]. Unfortunately, no such standard exists for eukaryotes other than fungi.

Here we present and validate the VESPA (Vertebrate Eukaryote endoSymbiont and Parasite Analysis) primers and optimized protocol for eukaryotic endosymbiont metabarcoding that resolves the issues described above. We compare VESPA to published methods in silico and using community standards comprised of cloned DNA from eukaryotic endosymbiont lineages across the Tree of Life. We then quantify off-target signal abundance and finally compare our protocol to the gold standard of microscopy using clinical samples. Our results show that VESPA and our community standard constitute a major advance that should enable microbiome-like insights into the structure and function of vertebrate-associated eukaryotic endosymbiont communities.

## Results

Here we compile and evaluate published methods for metabarcoding vertebrate-associated eukaryotic endosymbionts and choose a marker gene and region for amplification. We then compare the relevant subset of published methods to a protocol of our own design in a progressive series of experiments. We begin with in silico PCR, proceed to amplification of single parasite DNA templates, conduct metabarcoding using two engineered mock community standards, and assess off-target signals in fecal samples. We finally apply the best-performing protocol to clinical samples from humans and non-human primates and compare results to those obtained with microscopy.

### Methods review and protocol design

In a literature review consisting of 54 papers that used amplicon sequencing (metabarcoding) to characterize eukaryotic assemblages in vertebrate hosts (Supplementary Data 1), we identified eight marker genes, including nt-MD1 ($n = 1$), 12 S ($n = 1$), 28 S ($n = 1$), mitochondrial 16S ($n = 2$), mini-exon Tcl DTU ($n = 2$), CO1 ($n = 2$), ITS-2 ($n = 13$), and 18S ($n = 37$; Fig. 1a). Of these publications, 25 targeted specific sub-groups (e.g., nematodes or trypanosomes) and 29 used a pan-parasite/commensal approach. Based on the widespread incorporation of small subunit ribosomal RNA 18S gene (18S hereafter) sequences into databases, the standardized use of the counterpart prokaryotic 16S gene for bacterial metabarcoding, and evidence that non-protein coding genes outperform protein-coding genes as metabarcoding markers[36], we chose to pursue 18S as our marker gene.

18S contains hypervariable regions V1–V9, and the regions most commonly targeted in the studies reviewed were V4 ($n = 13$) and V9 ($n = 13$; Fig. 1b). The 18S V4 region has the highest entropy within the size limits of MiSeq v2 chemistry[37] and therefore the highest taxonomic resolution for this commonly used metabarcoding platform, so we chose to target this region. We identified a total of 22 published sets of V4 primers. Additionally, we created 18S V4 primers designed to target all vertebrate eukaryotic endosymbionts, consisting of 4 candidate forward primers and one reverse primer (see Methods section for details on primer design, Supplementary Table 1 for primer sequences and Fig. 1c for a map of primer binding sites).

### Testing metabarcoding methods for taxonomic coverage and resolution using in silico PCR

Testing all 22 published 18S V4 primer sets in silico (condensed for coverage analysis by combining primer sets in 2 cases for 19 total Primer IDs) yielded an average eukaryotic endosymbiont coverage of 64.9% (Table 1, bolded columns). No primer set recognized both *Plasmodium* and *Giardia*, and 7 sets did not recognize either (Table 1, final two columns). We found significant off-target coverage (>5%) of bacterial and/or archaeal groups for 4 of 22 sets, and the primer set with the highest overall eukaryotic coverage (96.3%; Hugerth 2014 "563/1132") also had the highest coverage of archaea and bacteria (47.9% and 72.0% respectively; Table 1). Primer sets with >5% off-target prokaryotic coverage were not analyzed further as prokaryotes constitute the vast majority of genetic material in gut flora and fecal samples[38].

In silico PCR including our four primer sets alongside the remaining 18 published 18S V4 sets (condensed in 2 cases, total 19 Primer IDs) yielded coverage data spanning a wide range (5.7–98.0%; Table 2). Across target groups (normalizing by eligible accessions), our designed primers had the highest mean percent coverage, at 95.2–96.8%, and also the best complementarity as evidenced by the lowest score in a rank sum analysis (Table 2, penultimate column). Overall coverage of fungal groups was high with all primer sets (45.1–94.1%, mean 86.2%; Table 2, final column), as was expected based on 18S sequence similarity between fungi and eukaryotic endosymbionts[39].

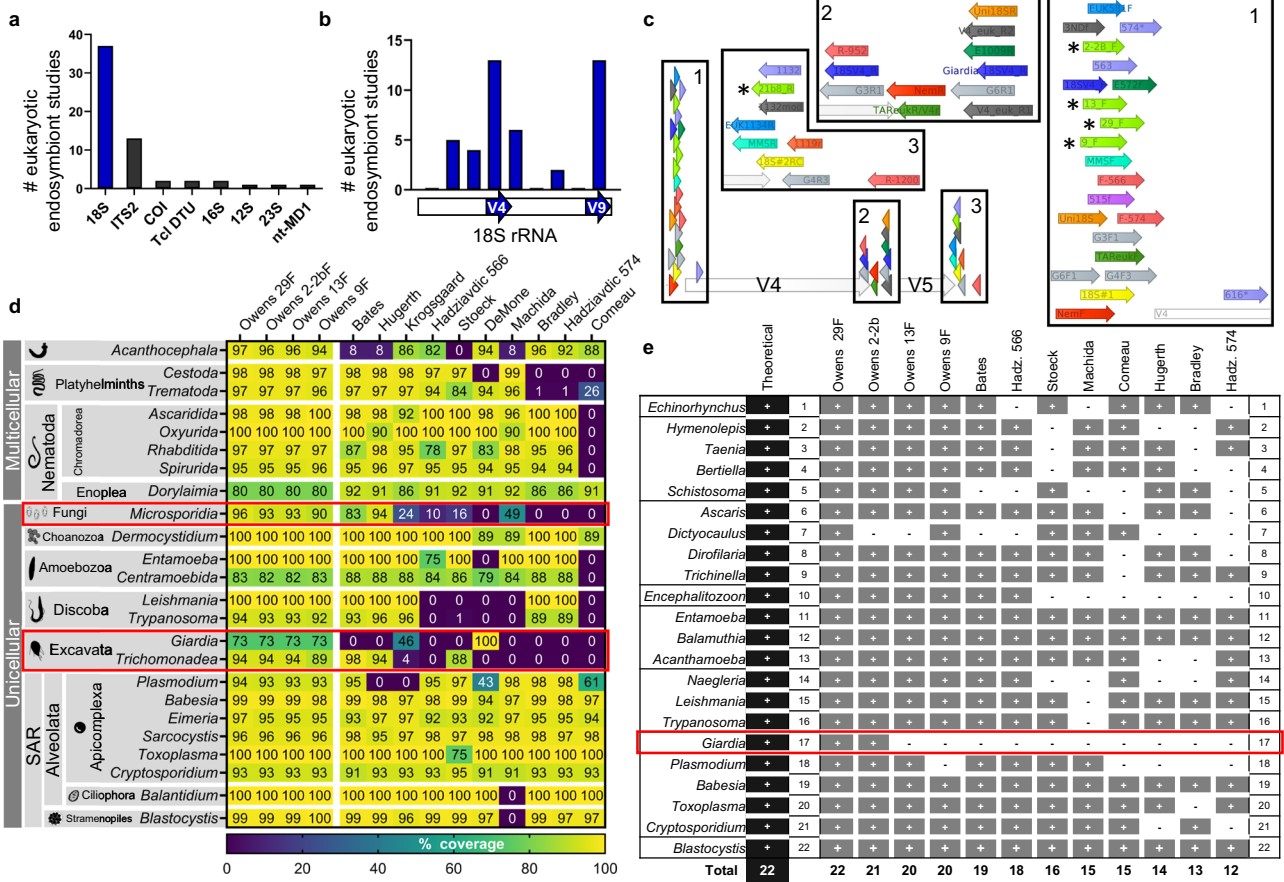

**Fig. 1 | VESPA development and evaluation. a** Histogram of marker genes identified in a literature review of 54 host-associated eukaryotic endosymbiont studies (see Supplementary Data 1). **b** 18S rRNA primer sets from our literature review shown as a histogram binned by location along the 18S gene. Hypervariable regions V4 and V9 are demarcated by blue arrows below the x-axis. **c** Generalized map of hypervariable regions V4–V5 (open arrows) of eukaryotic 18S SSU rRNA gene. Newly designed (asterisks) and published metabarcoding primer sets are shown as colored arrows and boxed areas 1–3 are expanded as insets. See Supplementary Table 1 for full primer names and sequences. **d** Heat map of published and new 18S V4 primer set coverage across clades exclusively containing eukaryotic endosymbionts of vertebrates. Percent overall complementarity (% coverage) is shown as numbers and as a color scale (color key below heatmap) with taxonomic labels to the left. Red boxes highlight clades with low overall (problematic) coverage. **e** Vertebrate eukaryotic endosymbiont PCR panel showing amplification (+) or lack of amplification (-) of single-organism gDNA templates across new and published primer sets. Total represents the number of successful amplifications per primer out of 22 possible, shown in left-most "Theoretical" column. Red box highlights clade with low overall (problematic) amplification.

In silico coverage analysis using finer-resolution groups (Fig. 1d) showed that our designed primers consistently amplified (defined as coverage of 50% or higher) all 24 clades of eukaryotes tested whereas no other primer sets did. Particularly problematic were *Giardia* (recognized by our primers and one other set in which a second reverse primer must be used to specifically amplify *Giardia*), *Microsporidia* (recognized by our primers and two other sets), and *Trichomonadea* (recognized by our primers and three other sets; Fig. 1d, red boxes).

We tested the taxonomic resolution of the 18S V4 region amplified by our primers by evaluating the amplicon sequences of known eukaryotic vertebrate endosymbionts generated by in silico PCR (*n* = 3769). Overall, 98.3% of sequences (*n* = 3704) could be resolved to the species level, 1.4% (*n* = 52) to the genus level, and 0.3% (*n* = 13) to the family level (Table 3). Normalized by total sequences, Platyhelminthes had the greatest number of sequences not resolved to the species or genus level, but with only 6 of 531 (1.13%) sequences assigned to more than one genus (Supplementary Table 2).

**Testing metabarcoding methods for on-target amplification using purified DNA**

In PCR amplification of genomic DNA (gDNA) from 22 individual eukaryotic endosymbiont organisms (Supplementary Table 3), all four

sets of candidate primers amplified more organisms than did any of the published primer sets (Owens 29F: 22 of 22, Owens 2-2bF: 21 of 22, Owens 13F: 20 of 22, Owens 9F: 20 of 22), followed by the Bates (19 of 22), Hadziavdic 566 (18 of 22), and Stoeck (16 of 22) sets (Fig. 1e). Furthermore, two of the new sets were the only primers to successfully amplify 18S V4 from *Giardia* gDNA (Owens 29 F and Owens 2-2bF), as expected based on in silico data (Fig. 1e, red box).

**Testing metabarcoding methods for amplification bias using a community standard**

Community standards are not available for eukaryotic endosymbionts, so we collected host-associated protozoa (*n* = 10), helminths (*n* = 5), and a microsporidian (*n* = 1) (Supplementary Table 3) from various sources (e.g., specimen repositories, veterinary post-mortem examinations). We then isolated 18S genes from these samples and combined them at different relative abundances to create two community standards (Fig. 2a), one consisting of equal 18S gene copies from each organism, which we named Equimolar EukMix, and one with unequal 18S gene copies following a logarithmic function (maximum 400-fold abundance difference), which we named Log EukMix (Supplementary Table 4). Metabarcoding Equimolar EukMix and Log EukMix with previously published and newly designed primers allowed us to directly compare empirical read abundances for each organism to

**Table 1 | In silico taxonomic coverage for published 18S V4 primer sets**

| Primer ID | Primer sets | Off-target groups | | Eukaryotic endosymbiont groups | | Specific examples | |
|---|---|---|---|---|---|---|---|
| | n = | 20,197 Archaea | 381,535 Bacteria | 4229 Helminths | 15,265 Protozoa | 198 *Plasmodium* | 23 *Giardia* |
| Bates[106] | 515 f/1119r | 0 | 0 | **80.4** | **95.9** | 94.8 | 0 |
| Bower[a][107] | 18SEUK581F/18SEUK1134R | 46.2[a] | 8.2[a] | **0.4** | **82.4** | 0 | 72.7 |
| Bradley[37] | TAReuk454FWD1/V4r | 0 | 0 | **48.9** | **67.1** | 97.9 | 0 |
| C-S[108]/Brate 2[109] | 3NDf/V4_euk_R2 | 0 | 0 | **50.8** | **22.8** | 0 | 0 |
| C-S[108]/Brate 1[109] | 3NDf/V4_euk_R1 | 0 | 0 | **5.8** | **21.1** | 0 | 0 |
| C-S[108]/Geisen[110] | 3NDf/1132mod | 0.3 | 0 | **80.7** | **94.2** | 0 | 0 |
| Comeau[111] | E572F/E1009R | 0 | 0 | **65.3** | **44.5** | 0 | 0 |
| DeMone[b][112] | 18SV4_F/_R/Giardia_R | 0 | 0 | **86.4** | **62.3** | 0 | 100 |
| Hadziavdic 566[73] | F-566/R-1200 | 0 | 0 | **76.4** | **81** | 99.6 | 0 |
| Hadziavdic 574[73] | F-574/R-952 | 0 | 0 | **48.3** | **62.9** | 61.3 | 0 |
| Hugerth 574[a][113] | 574/1132 | 12.5[a] | 0 | **80** | **94.2** | 0 | 0 |
| Hugerth 616[113] | 616/1132 | 3.3 | 0.2 | **93.1** | **75.8** | 0 | 45.5 |
| Hugerth 563[a][113] | 563/1132 | 47.9[a] | 72[a] | **96.1** | **96.4** | 0 | 100 |
| Krogsgaard[b][32] | G3F1/R1/G4F3/R3/G6F1/R1 | 0 | 0 | **78.5** | **67** | 94.8 | 0 |
| Machida[114] | 18S#1/18S#2RC | 0 | 0 | **78.1** | **45.2** | 97.9 | 0 |
| Sikder[a][115] | MMSF/MMSR | 17.5[a] | 0 | **79.3** | **42.7** | 0 | 0 |
| Stoeck[116] | TAReuk454F1/R3 | 0 | 0 | **49.1** | **78.4** | 97.9 | 0 |
| Wood[117] | Nem18SlongF/Nem18SlongR | 0 | 0 | **32.2** | **25.2** | 2.6 | 0 |
| Zhan[118] | Uni18S/R | 0 | 0 | **72.8** | **64** | 0 | 0 |

Bold font denotes target organisms. See Supplementary Table 1 for full primer names and sequences.
Numbers shown are % coverage allowing for 1 mismatch with a 2-base pair 3′ window using the SILVA 138.1 SSU rRNA NR Ref database.
n, number of total eligible accessions.
[a]Removed from further analysis due to high prokaryotic complementarity.
[b]Multiple primer sets were combined for analysis.

their predicted abundances (Fig. 2b, top panel) over a wide range of 18S gene abundances. Analysis of Equimolar EukMix and Log EukMix metabarcoding reads demonstrated that the mean fold difference from the theoretical input (a reflection of PCR bias[37]) was significantly lower with designed primer set Owens 29F (Fig. 2b, bottom panel) compared to all published primer sets (see Supplementary Table 5 for exact *P*-values). Similarly, overall abundance distributions resulting from primer set Owens 29 F were strongly positively correlated (a reflection of accurate community reconstruction[40,41]) to that of both mock community inputs (Equimolar EukMix: $r = 0.938$, Log EukMix: $r = 0.9554$; Pearson correlation coefficient) and more closely approximated the actual community than any other primer set tested (Fig. 2c). We therefore chose 29F/21b8R as the primer set for our finalized VESPA metabarcoding protocol (see Supplementary Note 1—VESPA protocol).

**Testing VESPA for off-target amplification of prokaryotic, fungal, and host sequences**
To test the performance and off-target amplification abundance of VESPA primers when applied to clinically-relevant samples, we used DNA extracted from whole human feces ($n = 40$) as input for metabarcoding, which yielded a mean 44,311 reads per sample after quality-filtering (range: 23,093–89,510; Fig. 3a). Reads originating from bacteria and archaea accounted for a mean 3.8% (range: 0–28.5%), host accounted for a mean 6.4% (range: 0–32.1%), and fungi accounted for a mean 3.1% (range: 0–29.7%; Fig. 3b). Reads originating from on-target organisms (Fig. 3a) accounted for an overall mean of 86.7% (range: 55.1–99.4%; Fig. 3b).

**VESPA compared to microscopy in human samples**
VESPA analysis of 12 human clinical samples yielded high-quality data (Supplementary Table 6) including low proportions of off-target

prokaryotic reads (Fig. 4a) and host reads (host read mean = 2.97% per sample, range: 0.11–17.4%) and correspondingly high proportions of endosymbiont reads (Fig. 4b, c).

VESPA successfully identified all three helminth and seven protozoan taxa identified with microscopy (Fig. 4d) and found these taxa in more individuals than did microscopy, with 61.4% of positive samples identified solely by VESPA (Fig. 4e). Conversely, no positives were identified by microscopy alone. Four additional taxa were found exclusively by VESPA, including one helminth, *Trichuris trichuria* (1 positive of 12 samples*)*, and three protozoa, *Entamoeba hartmanni* (10 positives of 12 samples*)*, *Enteromonas hominis* (3 positives of 12 samples), and *Pentatrichomonas hominis* (1 positive of 12 samples). Three of 12 patients were known by taxon-specific PCR to be infected with *Onchocerca*, which is not visible microscopically in feces, and all 3 were positive by VESPA. Overall, taxonomic richness was statistically significantly higher by VESPA than by microscopy for both helminths (mean richness = 0.5 by microscopy, 1.92 by VESPA, Wilcoxon matched-pairs signed rank test, 2-tailed, $P = 0.001$) and protozoa (mean richness = 2.33 by microscopy, 5.67 by VESPA, Wilcoxon matched-pairs signed rank test, 2-tailed, $P = 0.0005$; Fig. 4f, left panel). Prevalence was also higher by VESPA for helminths (mean prevalence = 0.25 by microscopy, 0.60 by VESPA, Wilcoxon matched-pairs signed rank test, 2-tailed, $P = 0.25$) and protozoa (mean prevalence = 0.23 by microscopy, 0.54 by VESPA, Wilcoxon matched-pairs signed rank test, 2-tailed, $P = 0.002$; Fig. 4f, right panel).

**VESPA compared to microscopy in non-human primate samples**
VESPA analysis of 40 non-human primate clinical samples yielded high-quality sequencing reads (Supplementary Table 6) with low proportions of off-target prokaryotic reads (Fig. 5a) and host sequence reads (host read mean = 3.2% per sample, range: 0%–18.49%) and correspondingly high proportions of endosymbiont reads (Fig. 5b, c).

**Table 2 | In silico taxonomic coverage of host-associated helminths/protozoa and fungi for published and newly designed 18S V4 primer sets**

| Primer ID | Mean | n = 3097 Helminths | n = 2913 Protozoa | Rank Helminths | Protozoa | Rank sum | n = 9373 Fungi |
|---|---|---|---|---|---|---|---|
| Owens 29F | **96.8%** | 95.5% | 98.0% | 1 | 1 | **2** | 91.1% |
| Owens 2-2b | **96.4%** | 94.9% | 97.9% | 2 | 2 | **4** | 92.2% |
| Owens 13F | **96.4%** | 94.9% | 97.9% | 2 | 2 | **4** | 92.4% |
| Owens 9F | **95.2%** | 94.4% | 96.0% | 4 | 4 | **8** | 92.4% |
| Bates | **88.2%** | 80.4% | 95.9% | 8 | 5 | **13** | 93.7% |
| Hugerth | **84.5%** | 93.1% | 75.8% | 5 | 8 | **13** | 94.1% |
| Krogsgaard[a] | **81.0%** | 93.0% | 69.0% | 6 | 9 | **15** | 91.8% |
| Hadziavdic 566 | **78.7%** | 76.4% | 81.0% | 10 | 6 | **16** | 91.4% |
| DeMone[a] | **75.3%** | 86.4% | 64.1% | 7 | 11 | **18** | 81.8% |
| Stoeck | **63.8%** | 49.1% | 78.4% | 13 | 7 | **20** | 79.4% |
| Machida | **61.7%** | 78.1% | 45.2% | 9 | 14 | **23** | 92.5% |
| Bradley | **58.0%** | 48.9% | 67.1% | 14 | 10 | **24** | 69.5% |
| Hadziavdic 574 | **55.6%** | 48.3% | 62.9% | 15 | 12 | **27** | 78.9% |
| Comeau | **54.9%** | 65.3% | 44.5% | 11 | 15 | **26** | 91.4% |
| C-S/Geisen[b] | **47.5%** | 40.7% | 54.2% | 16 | 13 | **29** | 93.0% |
| C-S/Brate 2[b] | **36.8%** | 50.8% | 22.8% | 12 | 17 | **29** | 90.0% |
| Wood[b] | **28.7%** | 32.2% | 25.2% | 17 | 16 | **33** | 45.1% |
| Zhan[b] | **14.0%** | 5.7% | 22.3% | 19 | 18 | **37** | 83.6% |
| C-S/Brate 1[b] | **13.5%** | 5.8% | 21.1% | 18 | 19 | **37** | 93.8% |

Bold font indicates overall metrics (mean and rank sum). See Supplementary Table 1 for full primer names and sequences.
The first four rows represent primers designed in this study.
%, % coverage calculated allowing for 1 mismatch with a 2-base pair 3′ window using the SILVA 138 SSU rRNA NR Ref database; n, number of eligible accessions; Mean, mean coverage of all parasite/commensal groups.
[a]Multiple primer sets combined for analysis.
[b]<50% overall mean target complementarity.

**Table 3 | In silico 18S V4 taxonomic resolution of host-associated protozoa and helminths**

| | Total unique sequences | Count 1 species | >1 species | >1 genus | Percent 1 species | >1 species | >1 genus |
|---|---|---|---|---|---|---|---|
| Blastocystis | 141 | 139 | 2 | 0 | 0.986 | 0.014 | 0.000 |
| Ciliophora | 772 | 770 | 2 | 0 | 0.997 | 0.003 | 0.000 |
| Apicomplexa | 1476 | 1458 | 17 | 1 | 0.988 | 0.012 | 0.001 |
| Amoebazoa | 317 | 316 | 1 | 0 | 0.997 | 0.003 | 0.000 |
| Acanthocephala | 72 | 69 | 3 | 0 | 0.958 | 0.042 | 0.000 |
| Platyhelminthes | 531 | 517 | 6 | 8 | 0.974 | 0.011 | 0.015 |
| Nematoda | 460 | 435 | 21 | 4 | 0.946 | 0.046 | 0.009 |

Third and sixth columns represent species-level resolution.
See Supplementary Table 2 for full list of unresolved species.

VESPA successfully identified all eight helminth and six protozoan taxa identified with microscopy (Fig. 5d) and found these taxa in more individuals than did microscopy, with 47.08% of positive samples identified by VESPA only (Fig. 5e). One positive out of 29 total for a helminth (*Physaloptera* sp. 1) and 2 positives out of 28 total for a protozoan (*Balantidium coli*) were identified by microscopy only. Six additional taxa were found exclusively by VESPA: *Entamoeba chattoni* (16 positives of 40 samples), *Endolimax nana* (19 positives of 40 samples), *Enteromonas* sp. (6 positives of 40 samples), *Piroplasmida* sp. (2 positives of 40 samples), *Blastocystis* sp. (38 positives of 40 samples), and *Enterocytozoon bieneusi* (3 positives of 40 samples; Fig. 5d, e). *Piroplasmida* are intraerythrocytic parasites not visible in fecal samples and were found in 2 of 40 samples with VESPA. Thirty-one samples were positive for the *Entamoeba histolytica/dispar* species complex by microscopy and the same 31 samples were found to be positive by

VESPA but could be further taxonomically resolved as *Entamoeba dispar* in all cases, constituting resolution of a cryptic species complex. Richness was higher by VESPA than by microscopy for helminths (mean richness = 1.73 by microscopy, 2.13 by VESPA, Wilcoxon matched-pairs signed rank test, 2-tailed, $P = 0.0009$), protozoa (mean richness = 2.8 by microscopy, 5.5 by VESPA, Wilcoxon matched-pairs signed rank test, 2-tailed, $P < 0.0001$), and microsporidia (mean richness = 0 by microscopy, 0.08 by VESPA, Wilcoxon matched-pairs signed rank test, 2-tailed, $P = 0.25$; Fig. 5f, left panel). Prevalence was also higher by VESPA than by microscopy for all three parasite groups (helminth mean prevalence = 0.22 by microscopy, 0.26 by VESPA, Wilcoxon matched-pairs signed rank test, 2-tailed, $P = 0.33$; protozoa mean prevalence = 0.22 by microscopy, 0.43 by VESPA, Wilcoxon matched-pairs signed rank test, 2-tailed, $P = 0.002$; microsporidia mean prevalence = 0 by microscopy, 0.8 by VESPA Fig. 5f, right panel).

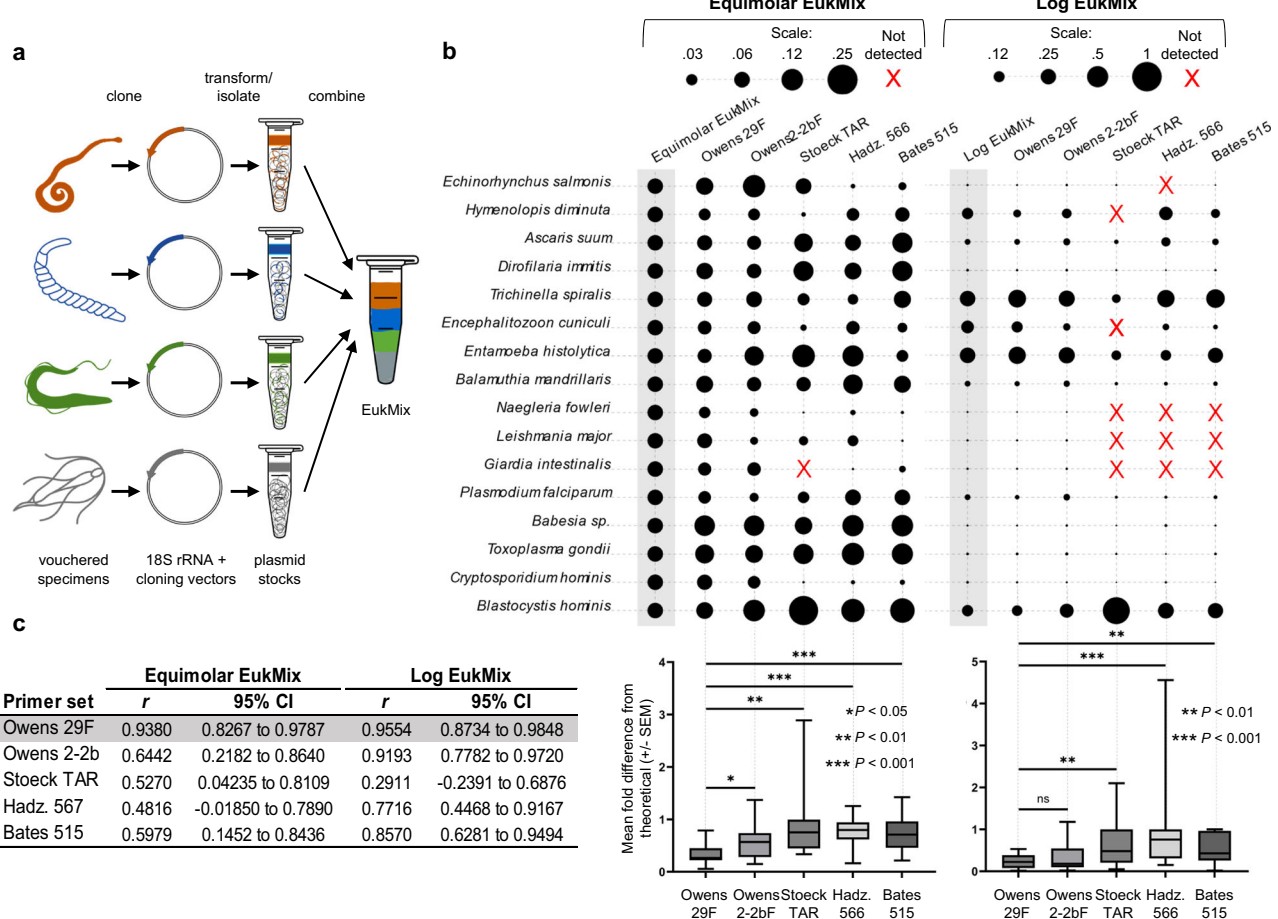

**Fig. 2 | Testing metabarcoding methods for amplification bias using community standards. a** Schematic overview of EukMix creation via 18S isolation and cloning. **b** Equimolar EukMix (left panel) and Log EukMix (right panel) community standard metabarcoding across primer sets (columns, labels at top and bottom) as compared to theoretical input (leftmost columns, gray boxes) shown as % abundance of reads per organism corresponding to bubble area. Any organism not detected (0 reads after quality filtering) is shown as a red X. See Supplementary Table 3 for parasite sources and strains. Bottom half of panels shows box plots of mean fold difference from theoretical input of three replicates ± standard error of

the mean (SEM) in which the central line is the median, box limits indicate first to third quartile range, and whiskers display minimum and maximum values. *P* values are derived from two-tailed Wilcoxon matched-pairs signed rank tests. ns, not significant. See Supplementary Table 5 for exact *P*-values. **c** Pearson correlation coefficients for the relationship between theoretical and observed read abundances for Equimolar EukMix (top) and Log EukMix (bottom). Primer set Owens 29F recovered the underlying communities most accurately (shading). Source data are provided as a Source Data file.

## Discussion

To identify a single protocol for the universal identification of vertebrate-associated eukaryotic endosymbionts in community assemblages, we analyzed published approaches and found a wide range of amplification targets and protocols. From this literature review, we chose to focus on the 18S V4 locus and designed primers to recognize all known groups of eukaryotic endosymbionts. We then tested published primers and our newly designed primers in a series of experiments in silico and in vitro to determine which protocols, if any, could accurately reconstruct eukaryotic endosymbiont communities. Our results clearly show that metabarcoding using the designed primer set 29F recognizes the greatest range of eukaryotic endosymbionts of interest with the least PCR bias of any method tested. When applied to DNA extracted from whole human fecal samples, metabarcoding using the 29F primer set resulted in low abundances of off-target prokaryotic, fungal, and host reads with the majority of reads assigned to eukaryotic endosymbionts of interest. We name our new primers and optimized protocol VESPA (Vertebrate Eukaryotic endoSymbiont and Parasite Analysis).

VESPA recognized more eukaryotic endosymbiont groups in silico than did other published methods tested, including methods that used

multiple primer sets to increase coverage. Multiple primer sets, usually involving multiple independent PCR amplifications, are a feasible strategy for increasing coverage[32,33]. However, this approach adds reagent costs and presents technical challenges related to sequencing and bioinformatics[42,43]. Our single primer set approach should therefore reduce barriers to entry for adopting our new protocol. We then corroborated these in silico results with amplification of purified targets and similarly found that our primer sets amplified the greatest range of single organisms in vitro.

To examine the performance of published methods and VESPA, we directly compared assays by using two mock community standards, Equimolar EukMix and Log EukMix, as input for metabarcoding. In both cases, results from VESPA reflected the underlying composition of the community standard more accurately than did results from other assays. The two EukMix community standards should be useful for quality control in laboratories choosing to adopt our protocol, and for standardization and validation, much as community standards containing bacteria and fungi have enabled standardization of microbiome protocols[35,44]. We note that the relationship between sequencing reads and organism abundance or biomass is complicated by wide variation in 18S copy number among eukaryotic endosymbionts[34,45].

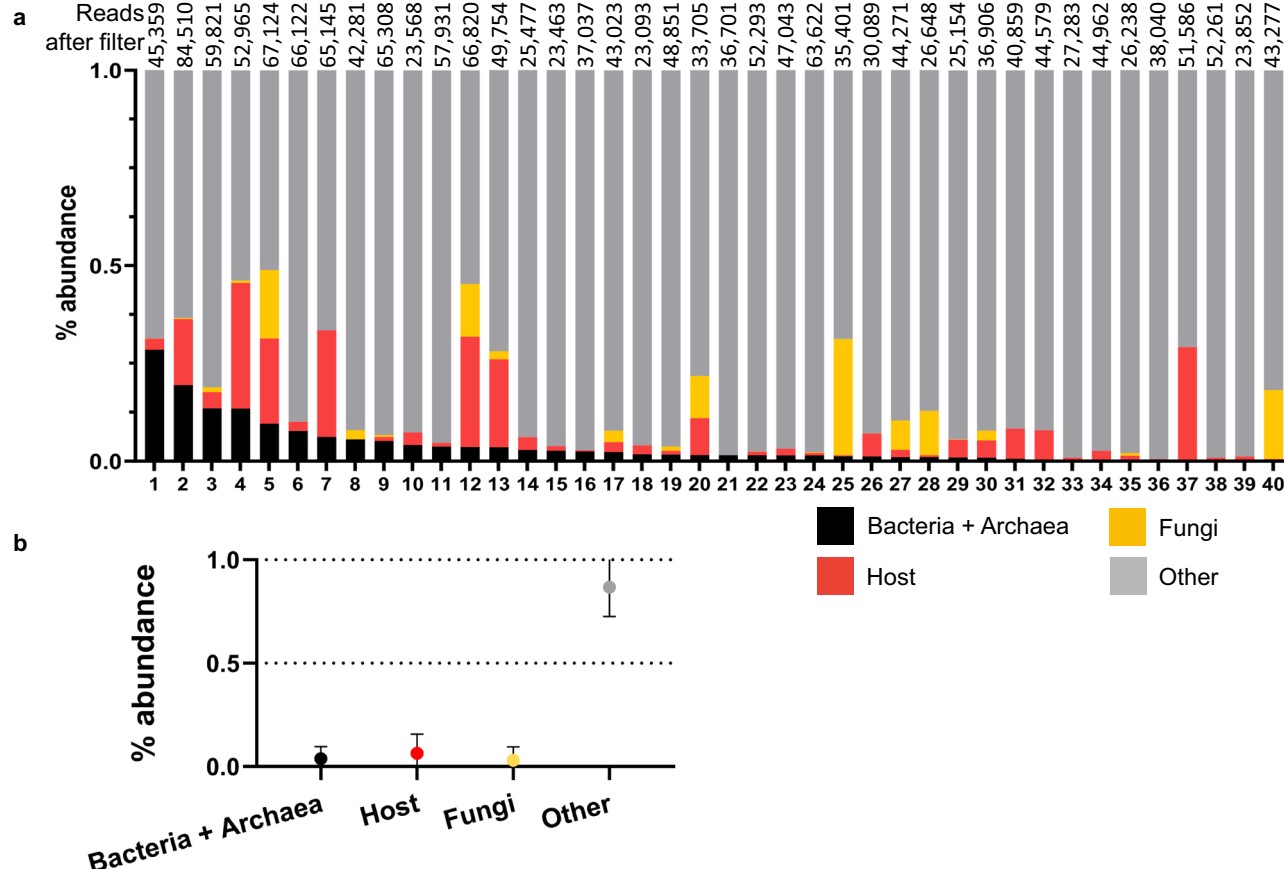

**Fig. 3 | Testing VESPA for off-target amplification with fecal samples. a** VESPA metabarcoding data shown as percent relative abundance for each organism category. Off-target reads are Bacteria + Archaea, Host, and Fungi. On-target reads are Other. Read numbers after quality filter ($Q = 30$) for each sample are above the bars. **b** Relative abundance of each organism category from (**a**) ($n = 40$ independent samples) displayed as mean ± SEM. Source data are provided as a Source Data file.

Copy number corrections have been applied in studies of other systems[46,47], and such corrections could prove useful for investigations where quantifying organism abundance or biomass are the desired outputs.

We also note that all marker genes have limits of taxonomic resolution[48]. Use of 18S for metabarcoding affords lower resolution than other, less evolutionarily-conserved marker genes such as mitochondrial cytochrome oxidase 1 (CO1)[49,50], although several commonly cited studies only examined hypervariable regions V1 – V2 or V9[51,52]. Although 18S V4 underperforms compared to CO1 in some systems[53,54], recent studies of marine invertebrates have demonstrated that 18S V4 can have comparable species-level resolution to CO1[55-60] and may provide more reliable estimates of biomass[61]. Indeed, we found that 98.3% of sequences could be resolved to the species level using our 18S V4 primers. This analysis does not, however, take into account the stochastic effects of metabarcoding in complex samples or the fact that species-level resolution of every eukaryotic endosymbiont group is not possible using 18S[57]. There is an inherent tradeoff between coverage and resolution[62] in marker gene selection that in fact underlies our choice of the 18S gene[63], with the understanding that some identifications may only be genus level.

When VESPA was applied to human fecal samples to assess on-target vs off-target amplification, a mean 86.7% of resulting reads were identified as on-target reads of interest (Fig. 2d). Overall amplification of fungi was low (mean 3.1%; Fig. 2d), despite high primer cross-complementarity with fungal 18S sequences (Table 2). We suspect this finding to have resulted from the low relative amount of fungal organisms in human feces compared to bacteria and archaea[64]. Fecal sample composition varies with factors such as host species[65], diet[66], age[67], and immune status[68], thus we predict that fungal read abundance may be higher in some individuals and some hosts, which may explain the higher percentages of fungal reads in non-human primate samples (Fig. 5) than in human samples.

Compared to microscopic examination, VESPA detected protozoa, microsporidia, and helminths in more individuals, identified additional organisms, differentiated the morphologically identical organisms *Entamoeba histolytica* from *E. dispar*, and identified organisms not visible in fecal samples. We suspect that the greater sensitivity of VESPA results from the nature of molecular amplification −namely, that PCR can detect a theoretical minimum of one molecule of target DNA[69]. Microscopy-negative samples that were PCR positive by VESPA may not have contained intact organisms or their eggs or may even have been positive by virtue of the presence of small amounts of cell-free DNA[70]. In this light, we caution that our protocol will likely be most useful for applications where the presence of eukaryotic endosymbiont DNA is itself taxonomically informative, regardless of whether that DNA represents an intact or viable organism.

Because of the labor-intensive nature of microscopy and its dependence on trained experts, VESPA will also be useful for studies which are large-scale or performed in multiple laboratories, where labor costs and inter-observer variability would otherwise be impractical. In this light, we note that microscopy identified three positive samples not identified by VESPA in non-human primates. We suspect that these findings may represent microscopy false positives, especially because these two taxa (*Physaloptera* and *Balantidium*) are notoriously difficult to identify morphologically[71,72].

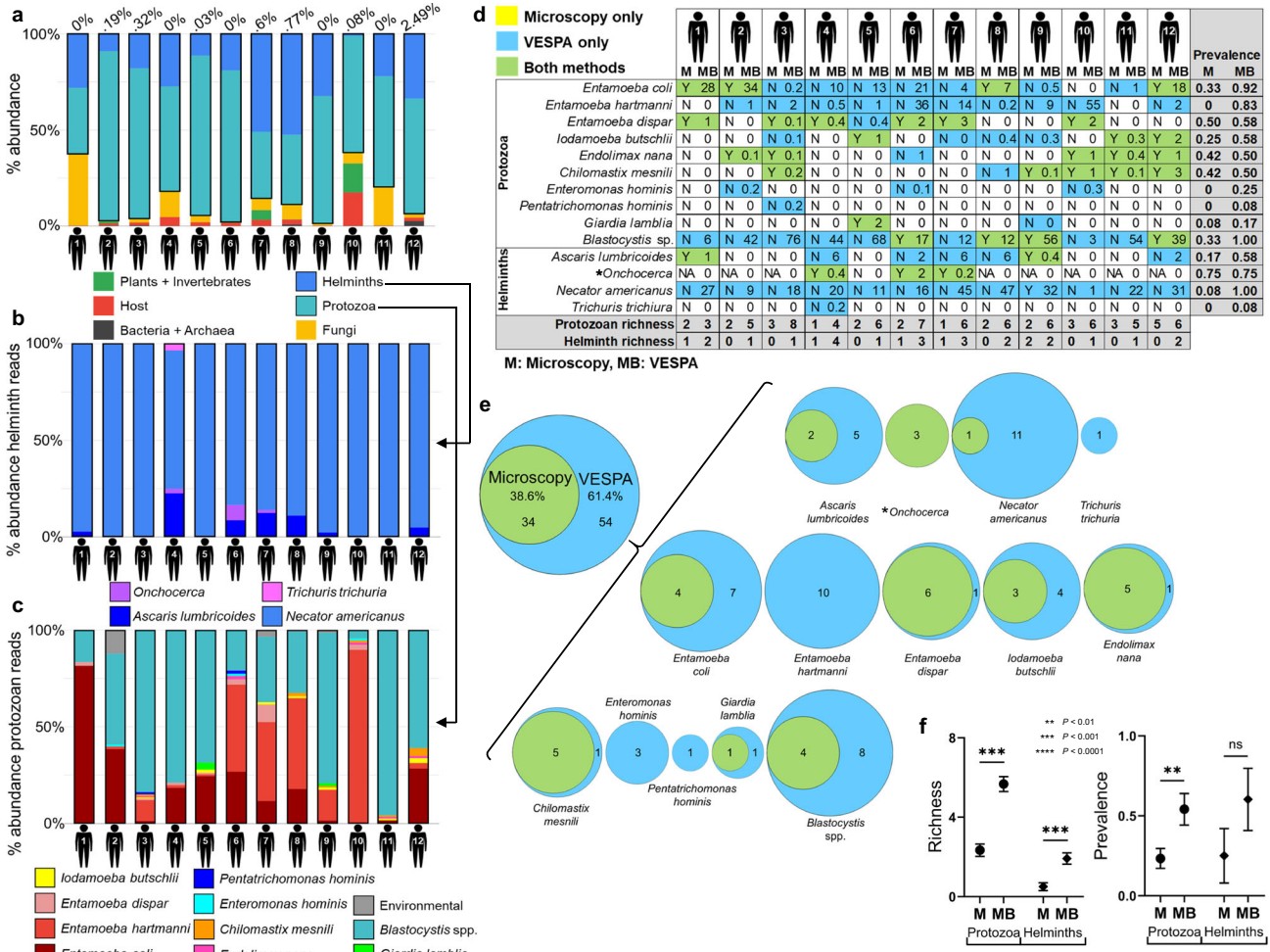

**Fig. 4 | VESPA compared to microscopy in human clinical samples.** VESPA metabarcoding data. VESPA data are shown as percent relative abundance of each organism category with (**a**) all quality-filtered reads included, (**b**) with helminth reads only, or (**c**) with protozoal reads only (archaea, bacteria, host, plants, invertebrates, and fungi removed in (**b**, **c**)). **a** Numbers above bars are the total percentage of prokaryotic (bacterial + archaeal) reads. **d** Microscopy versus VESPA. Microscopy findings (M) are shown as a presence/absence (Y = present, N = absent, NA = not assessed) and VESPA metabarcoding (MB) findings are shown as % abundance of quality-filtered reads. Blue cells represent detection by VESPA, green cells by both VESPA and microscopy, and white cells by neither method. No organisms were identified by microscopy alone. Richness (final two rows, shaded cells) is defined as the total number of species detected by the specified method. Prevalence (final two columns, shaded cells) is defined as the proportion of the

population positive for an organism by the specified method. Note that *Onchocerca* is not detectable in fecal samples by microscopy (asterisk). **e** Proportional Venn diagrams of findings by microscopy versus VESPA. Individuals identified as positive for the listed organisms by VESPA (blue) or both (green) are shown as numbers in each circle. Overall findings summed over all organisms are shown to the left of the bracket (not to scale). Note that *Onchocerca* is not detectable in fecal samples by microscopy (asterisk). **f** Richness and prevalence calculations ($n = 12$ independent samples) for microscopy (M) and VESPA metabarcoding (MB) findings. Data are shown as mean ± SEM. Protozoa richness: $P = 0.0005$, Helminths richness: $P = 0.001$, Protozoa prevalence: $P = 0.002$, Helminths prevalence: $P = 0.25$. $P$ values are derived from Wilcoxon matched-pairs signed rank tests, 2-tailed. ns, not significant. Source data are provided as a Source Data file.

Our contribution with this work is a publicly available protocol for metabarcoding eukaryotic endosymbiont communities that outperforms published methods by all measures examined. VESPA is intentionally designed to have broad applicability, from microbial ecology to parasitology to clinical diagnostics. Although we tested VESPA using Illumina sequencing technology, it should be readily adaptable to other amplicon sequencing technologies available now and in the future. VESPA is compatible with existing bacterial and fungal pipelines, with metabarcoding of all three taxa run on the same sequencing platform. Addition of VESPA to established protocols for characterizing bacterial microbiomes and mycobiomes could have far reaching benefits. For example, it has been suggested that studies of the human gut microbiome should routinely incorporate analyses of eukaryotic diversity in order to capture overall microbial community function[5]. VESPA can provide this missing eukaryotic component and thereby enable cross-kingdom characterization of microbial ecosystem

structure and function, opening new avenues for basic and applied research.

## Methods

### Ethical statement
This research complies with all relevant ethical regulations. Clinical samples used in this study were excess material from concluded research and no new samples were collected for this study. Human fecal samples had been collected with appropriate IRB approval (Venezuelan Institute of Scientific Research IRB #DIR-0609/1542/2015 and The University of Wisconsin Madison IRB #2013-1463), including written informed consent in all cases. Non-human primate fecal samples had been collected with appropriate IACUC approval (The University of Wisconsin-Madison's IACUC protocol #V1490). All samples had been completely de-identified prior to use and no demographic or identifying information such as age or sex/gender were provided.

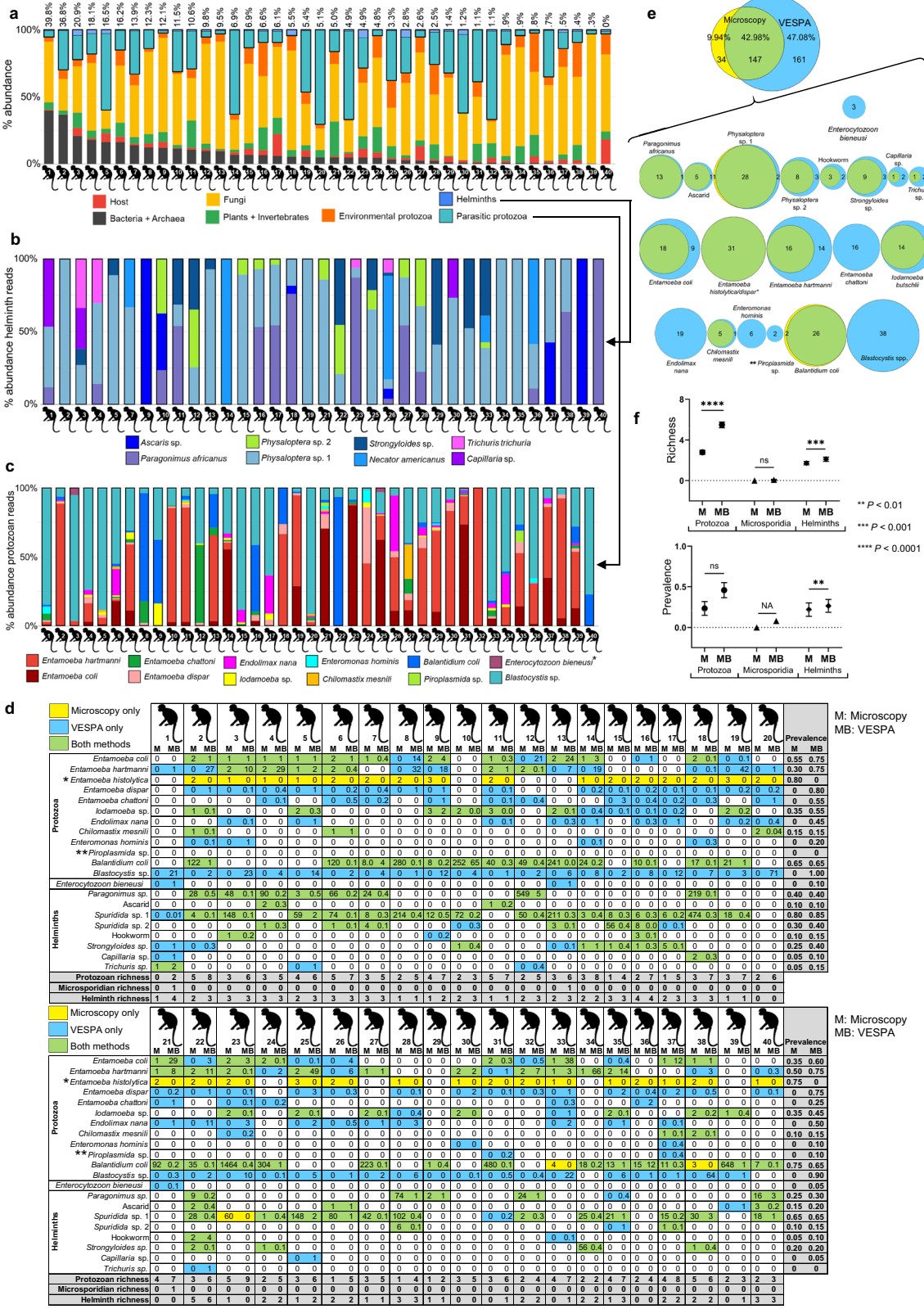

## Methods review and new protocol design

Literature searches were performed in January 2021 and updated in January 2023. Search terms or combinations of search terms including "Metagenomics," "Metagenomic barcoding," "Metabarcoding," "Targeted amplicon deep sequencing," "Eukaryotic microbiome," "Gastrointestinal," "Gut," "Parasite," "18S", "short sub unit", "short subunit", "short sub-unit", and "SSU" were used to query PubMed, Web of Science, and Google Scholar. Results were manually evaluated for relevance and details were compiled in an excel spreadsheet (Supplementary Data 1). We identified 96 studies including reviews and methods papers, 54 of which were primary research on vertebrate-associated eukaryotes. We chose to focus on 18S because in previous metabarcoding studies, non-coding genes outperformed coding genes[36,49], 18S has islands of conserved sequence interspersed with

**Fig. 5 | VESPA compared to microscopy in non-human primate clinical samples.** VESPA metabarcoding data. VESPA data are shown as percent relative abundance of each organism category with (**a**) all quality-filtered reads included, (**b**) with helminth reads only, or (**c**) with protozoal reads only (archaea, bacteria, host, plants, invertebrates, and fungi removed in (**b**, **c**)). **a** Numbers above bars are the total percentage of prokaryotic (bacterial + archaeal) reads. **c** Asterisk indicates a microsporidian parasite. **d** Microscopy versus VESPA. Microscopy findings (M) are shown as a qualitative score (1 least–3 most) for protozoa, larvae/gram feces for *Strongyloides*, and eggs/gram feces for all other helminths. VESPA findings (MB) are shown as % abundance of quality-filtered reads. Yellow cells represent parasite detection by microscopy, blue cells by VESPA, green cells by both methods, and white cells by neither method. Richness (final 2 rows, shaded cells) is defined as the total number of species detected by the specified method. Prevalence (final 2 columns, shaded cells) is defined as the proportion of the population positive for an organism by the specified method. Note that *Entamoeba histolytica* and *Entamoeba dispar* are a morphologic cryptic species complex that cannot be resolved by

microscopy (asterisk) and *Piroplasmida* sp. are not detectable in fecal samples by microscopy (double asterisk). **e** Proportional Venn diagrams of findings by microscopy versus VESPA. Individuals identified as positive for the listed organisms by microscopy (yellow), VESPA (blue), or both (green) are shown as numbers in each circle. Overall findings summed over all organisms are shown above the bracket (not to scale). Note that *Entamoeba histolytica* and *Entamoeba dispar* are a cryptic species complex that cannot be resolved by microscopy (asterisk) and *Piroplasmida* sp. are not detectable in fecal samples by microscopy (double asterisk). **f** Richness and prevalence calculations ($n = 40$ independent samples) for microscopy (M) and VESPA (MB) findings. Data are shown as mean ± SEM. Protozoa richness: $P < 0.0001$, Microsporidian richness: $P = 0.25$, Helminths richness: $P = 0.0009$, *Protozoa* prevalence: $P = 0.0586$, Helminths prevalence: $P = 0.0078$. $P$ values are derived from Wilcoxon matched-pairs signed rank tests, 2-tailed. ns, not significant. NA, not applicable (single data point only). Source data are provided as a Source Data file.

areas of high entropy (hypervariable regions), allowing broad priming for coverage and diverse amplicons for resolution[73], and database coverage for 18S is higher than for other loci[74]. Of the 9 hypervariable 18S regions, V4 has the highest taxonomic resolution[37], so we focused on this region and identified 22 sets of published V4 primers (Supplementary Table 1).

We also designed new 18S V4 primers with the goal of amplifying all eukaryotic endosymbiont groups with little to no prokaryotic complementarity. We began by creating a database of parasite/commensal 18S rRNA sequences containing representatives from all phylogenetic lineages containing at least one vertebrate-associated eukaryotic endosymbiont. We downloaded sequences from all known groups of endoparasites/endosymbionts from NCBI Genbank[75] or the SILVA 138.1 Small Subunit rRNA Non-Redundant Reference Database ($n = 510,508$ total accessions;[74,76] SILVA Ref NR hereafter) at a depth of one species per genus, beginning with the Centers for Disease Control's "Alphabetical Index of Parasitic Diseases"[77]. To ensure broad coverage of commensals, zoonoses, and novel organisms we added non-pathogenic protozoans of humans[78], parasites/commensals of great apes[79], and parasites of veterinary importance[80]. We then used MUSCLE[81] implemented in MEGA 11[82] to align the resulting 658 full-length 18S sequences, which covered a broad range of pathogenicity, vertebrate hosts, and tissue tropisms. To identify candidate conserved regions, we utilized the Arb software suite[83] and the ecoPrimers function in OBItools[84], with manual inspection and adjustment as needed. We then extracted every 16–20-mer candidate sequence within those regions and tested them for taxonomic coverage against SILVA Ref NR using the SILVA TestProbe and TestPrime tools[85]. Candidate primers with high overall complementarity were manually adjusted for maximum coverage.

We aimed to avoid degeneracy as it has been shown to create bias in 18S V4 amplification[37] and succeeded in the forward primer. Degeneracy was required in the reverse primer, although not in the four terminal 3′ nucleotides. Furthermore, of the three degenerate positions in the reverse primer, no targeted groups required all three degeneracies, and most required just one. To increase homogeneity and avoid potential biases against rare sequences, we used 5-deoxyinosine in the four-fold degenerate position instead of N, thereby limiting our reverse primer mixture to four distinct oligonucleotides[86].

The forward region identified for priming had higher GC content than the reverse region, so we forewent the standard guidelines for GC content and melting temperature differences in order to prioritize coverage, with the knowledge that we could later add Locked Nucleic Acids (LNAs) to modify the melting temperature if needed[87]. In the end, this modification was not necessary because the DNA polymerase for PCR (described below) tolerates a wide melting temperature range and has a universal annealing temperature regardless of primer

sequence. In total we designed 4 forward primers and one reverse primer (Supplementary Table 1) for further testing.

## Testing metabarcoding methods for taxonomic coverage and resolution using in silico PCR

For the initial analysis of published protocols for taxonomic coverage, we used locus-specific sequences (i.e., not including linkers, adapters, or barcode elements) from all 22 18S V4 primer sets identified in our literature search (Supplementary Data 1, Supplementary Table 1). In silico PCR of SILVA Ref NR was performed using the TestPrime tool allowing for a single mismatch and a mismatch-free two base pair 3′ window. For this analysis, helminth accessions included Acanthocephala ($n = 66$), Nematoda ($n = 2170$), and Platyhelminthes ($n = 1993$) and protozoa accessions included Amoebozoa ($n = 1148$), Discoba ($n = 1032$), Excavata ($n = 389$), Alveolata ($n = 9140$), and Stramenopiles ($n = 3556$). In two cases where multiple primer sets were used in combination (Krosgaard - three sets and DeMone-two sets), we tested each set individually and conservatively estimated coverage by reporting only the highest percentage for each taxon. Primer sets with >5% coverage of off-target prokaryote groups (archaea and bacteria) were not analyzed further ($n = 4$ sets).

In silico PCR was then used to evaluate the published primer sets remaining ($n = 18$) alongside our new candidate primers ($n = 4$; Supplementary Table 1). At this stage, we filtered target sequences to contain only parasites of vertebrates because the inclusion of environmental/free-living organisms can distort parasite coverage metrics. Specifically, we split clades that contained both free-living organisms and parasites of invertebrate hosts (e.g., *Rhabditida* and *Entamoeba*) into higher-resolution, curated groups. We included free-living, opportunistic parasites of clinical importance, including *Balamuthia mandrillaris* and *Naegleria fowleri*, and we excluded sequences whose label in the SILVA database was incorrect (i.e., the taxonomy string associated with the record did not match the phylogenetic placement in the guide tree; $n = 14$). Coverage metrics were normalized to eligible accession numbers, which were similar across primer sets because of similar priming locations in the V4 region (see Fig. 1c for primer map). We compared taxonomic coverage for primer sets using the TestPrime tool[85] and SILVA Ref NR[74,76] allowing for a single mismatch with a mismatch-free two base pair 3′ window. Primers with ≤50% overall mean coverage of target groups and methods that required more than a single primer set were not considered further.

Taxonomic resolution of the 18S V4 region amplified by our primer sets was assessed by running the TaxMan server[88] with the Owens 29F primer set as input, no sequence region targeting, a 5% mismatch allowance, a 3 bp 3′ mismatch window, and all other parameters set to default values. The database used was SILVA SSU NR rRNA v.128[74], and data were exported in non-redundant FASTA format, in which all headers contain full taxonomic identifiers and redundant sequences

(100% sequence identity) are concatenated. Sequences corresponding to vertebrate eukaryotic endosymbionts belonging to *Blastocystis*, Ciliophora, Apicomplexa, Amoebazoa, Acanthocephala, Platyhelminthes, and Nematoda were retrieved and binned at the species level (single species in taxonomic header ID), the genus level (more than 1 species of the same genus in taxonomic header ID), or the family level (more than one genus in the taxonomic header ID). All unique sequences with more than one species ID (i.e., sequences that could not be resolved to the species level) are shown in Supplementary Table 2.

### Testing metabarcoding methods for on-target amplification using purified DNA

We assessed amplification success of the remaining four designed and eight published primer sets across parasite groups using 22 gDNA isolates from single eukaryotic endoparasites as templates for PCR. For single organisms used for DNA extraction (parasitic worms and protozoa), samples (whole adult worms, cysts, proglottids, axenic cultures, or purified DNA) were obtained from expert parasitologists or from reputable reagent repositories (for sample details including sources see Supplementary Table 3). gDNA from whole worms and pelleted protozoal cultures were extracted using the DNeasy Blood and Tissue Kit (Qiagen, Hilden, Germany) using 0.2 g of starting material, eluted in Qiagen buffer AE, and stored at −20 °C. PCR conditions were as follows: 1 X Platinum II Hot Start PCR MasterMix (ThermoFisher, Waltham, Massachusetts, USA), 0.2 μM forward primer with Nextera adapter, 0.2 μM reverse primer with Nextera adapter, ThermoFisher 0.2 X Platinum II GC Enhancer, 0.8 ng/μl gDNA in a total 12.5 μl reaction; 94 °C for 2 min, 30 cycles of [94 °C for 15 s, 60 °C for 15 s, 68 °C for 15 s], and hold at 4 °C. Products were electrophoresed on a 1.5% agarose gel with SYBR gold DNA dye (ThermoFisher) and a 1 kb DNA size standard. Amplification was scored by band presence on an agarose gel upon visualization under UV illumination with a GelDoc XR imager (BioRad, Hercules, California, USA).

### Testing metabarcoding methods for amplification bias using a community standard

Preliminary metabarcoding experiments using mixes of gDNA from single parasites demonstrated a non-linear relationship between DNA input and sequence read abundance, likely due to rRNA copy number variation[89]. We addressed this issue by extracting, amplifying, and cloning parasite DNA from 16 vouchered parasite specimens from verified sources or identified by experts. 18S rRNA sequences were amplified with full-length universal or group-specific primers (see Supplementary Tables 1, 4) using Qiagen HotStar Plus Taq DNA polymerase according to manufacturer's instructions. Products were verified for size on an agarose gel and Sanger sequenced. Correct 18S sequences were cloned into a pCR4-TOPO vector using a TOPO TA Cloning Kit for Sequencing (Invitrogen, Waltham, Massachusetts, USA) and Invitrogen One Shot competent cells according to manufacturer's instructions. Colonies were screened by PCR and Sanger sequencing. Plasmid DNA (plDNA) extracted from verified transformants was mixed at equimolar ratios to create the equimolar EukMix community standard reagent (Supplementary Table 4). This strategy assures equal 18S copy number input among organisms, which, in the case of amplicon sequencing, enables assessment of primer bias and potential of the assays to yield quantitative data[90]. The actual distribution of abundances of eukaryotic endosymbionts in vertebrate hosts, however, is not even[91]. For example, mathematical models indicate that communities of vertebrate gut parasites are best represented by a logarithmic function[92] with a fold difference of up to ~150 between the most abundant and least abundant species[93]. Staggered communities have also been shown to test the dynamic range and detection limits of high-throughput assays better than even communities[94,95]. We

therefore created a staggered mock community with concentrations of 18S gene copies following a logarithmic function (Log EukMix) with the same 16 organisms as the Equimolar EukMix, with a maximum 400-fold difference among organism 18S abundances (Supplementary Table 4).

Metabarcoding using new and published primer sets was performed in triplicate with Equimolar EukMix and Log EukMix community standards as starting material using the procedure described below. Resulting sequencing reads were filtered for quality using a cutoff of $Q = 30$ and mapped to a database containing full-length 18S sequences of clones comprising the EukMix mock communities using a mapping stringency of 99% similarity and 99% length fraction in CLC genomics workbench v.10.2 (Qiagen). The resulting abundances for each community standard component were used to calculate Pearson Correlation Coefficients in R v.3.6.3, and GraphPad Prism v.8.4.3 was used for graphing data and for statistical analyses. The bubble plot representing percent abundances of all EukMix components was created with the bubble.pl script[96].

### Testing VESPA for off-target amplification of prokaryotic, fungal, and host sequences

In addition to containing eukaryotic endosymbionts of interest, fecal samples contain bacteria, fungi, and host material[38,64], and issues with abundant off-target reads originating from these organisms have been reported with 18S primers[97,98]. Many 18S primer sets have particularly high coverage of fungi due to the broad targeting of eukaryotes (Table 2, final column). We assessed our VESPA primers and metabarcoding protocol for amplification of off-target organisms (i.e., bacteria, archaea, fungi, and host) using DNA extracted from whole, unprocessed human fecal samples from western Uganda that were excess material collected as part of a concluded study[99] ($n = 40$). Appropriate IRB approval (The University of Wisconsin Madison IRB #2013-1463) was obtained by collaborators prior to collection and all samples were completely de-identified before use. Samples stored in 1:1 RNAlater nucleic acid preservation solution (ThermoFisher): fecal material by volume were thawed on ice and homogenized by vortexing. 0.2 g of homogenate was added to bead beating tubes (for a total of 0.1 g fecal material per extraction) and processed using the Qiagen DNeasy PowerLyzer PowerSoil kit according to manufacturer's recommendations. gDNA was eluted in Qiagen C6 buffer and stored at −20 °C. VESPA metabarcoding and data analysis were performed as described below.

### VESPA compared to microscopy

**Sample collection.** Clinical samples used in this work were excess material from concluded studies that had been previously evaluated for eukaryotic endosymbionts using microscopy. Human fecal samples had been collected from communities in Bolivar State, on the southern Venezuelan border with Brazil ($n = 12$). Non-human primate fecal samples had been collected from semi-free ranging Nigerian red capped mangabeys (*Cercocebus torquatus*) in a sanctuary[100] ($n = 40$). Appropriate IRB approvals (IVIC IRB #DIR-0609/1542/2015) and IACUC protocols (The University of Wisconsin-Madison's IACUC protocol # V1490) were obtained by each collaborator and all samples were completely de-identified prior to use. No demographic or identifying information was provided.

**Microscopy.** Microscopic analyses of non-human primate and human feces were performed as previously described[101]. Briefly, one gram of formalin preserved feces was concentrated via formalin-ethyl acetate sedimentation[100] and the sediment was examined in its entirety at ×10 objective light magnification for gastrointestinal parasites by an expert parasitologist. Additionally, one drop of sediment from each sample was examined at ×40 objective light magnification for identification of protozoa.

**Genomic DNA isolation.** Human fecal samples were processed to remove bacteria and debris as previously described[102]. Note that processing fecal samples in this manner is not necessary for VESPA metabarcoding and that we recommend use of whole fecal samples in our VESPA protocol (see Supplementary Note 1–VESPA protocol). Due to the unique and valuable nature of these samples (from a remote population known to harbor a markedly diverse gut eukaryotic community and previously examined by expert parasitologists) and lack of access to whole feces from this population, we used these samples for metabarcoding while acknowledging that the methodology may not be ideal for our intended applications. Briefly, feces were diluted in PBS (0.2 $M$ phosphate-buffered saline, pH 7.2), homogenized, filtered through sterile four-ply cotton gauze, pelleted for 5 min at 300 x $g$, resuspended in molecular grade water and layered on top of a 1.5 M sucrose solution. After centrifugation for 10 min at 1700 x $g$ the interphase was collected, and the process was repeated with a 0.75 M sucrose gradient. The resulting pellet was collected, washed in PBS, and resuspended in 2 ml of molecular-grade water. 0.2 ml of the resulting sample was used as starting material for phenol: chloroform: isoamyl alcohol (25: 24: 1) DNA extraction, eluted in IDTE buffer and stored at −20 °C.

Non-human primate fecal samples in 1:1 RNAlater nucleic acid preservation solution (ThermoFisher) were thawed on ice and homogenized by vortexing prior to transferring 0.2 g of homogenate to bead beating tubes (for a total of 0.1 g fecal material) for extraction using the Qiagen DNeasy PowerLyzer PowerSoil kit. gDNA was eluted in Qiagen C6 buffer and stored at −20 °C.

**Metabarcoding.** See Supplementary Note 1–VESPA protocol for step-by-step instructions. For compatibility of sequencing libraries across primer sets and amplicon library types, we created a 2-step Illumina Nextera-based protocol that does not require custom sequencing primers to be added to the sequencing cartridge. Primers for the first (amplicon) PCR consist of locus-specific sequences (Supplementary Table 1 for locus-specific primer sequences) with Nextera adapter sequences added to the 5′ end: F-TCGTCGGCAGCGTCAGAT GTGTATAAGAGACAG and R- GTCTCGTGGGCTCGGAGATGTGTA-TAAGAGACAG. A second, limited cycle (indexing) PCR was then used to add Nextera indexing primers to both ends. Note that Platinum II MasterMix (ThermoFisher) has a universal annealing temperature of 60 °C regardless of primer melting temperature. PCRs were run in triplicate with the following conditions: ThermoFisher 1 X Platinum II Hot Start PCR MasterMix, 0.2 μM forward primer with Nextera adapter, 0.2 μM reverse primer with Nextera adapter, 0.2 X ThermoFisher Platinum II GC Enhancer, 0.8 ng/μl gDNA in a total 12.5 μl reaction; 94 °C for 2 min, 30 cycles of [94 °C for 15 s, 60 °C for 15 s, 68 °C for 15 s], and hold at 4 °C. Triplicate reactions were then pooled and amplicons were cleaned using Ampure XP beads (Beckman Coulter, Brea, California, USA) then used as template for indexing PCR as follows: 1 X KAPA HiFi Hot Start ReadyMix (Roche, Basel, Switzerland), 1 X Nextera Unique Dual Index primers (Illumina, San Diego, California, USA), 1 μl of clean amplicons in a total 12.5 μl reaction; 95 °C for 3 min, 10 cycles of [95 °C for 30 s, 55 °C for 30 s, 72 °C for 30 s], 72 °C for 5 min, and hold at 4 °C. Indexed libraries were cleaned using Ampure XP beads (Beckman Coulter) assessed for concentration on a Qubit fluorometer (ThermoFisher), and pooled for sequencing on an Illumina MiSeq with 300 ×300 cycle chemistry using default index and sequencing read primers and 10–20% PhiX.

**Data processing and Bioinformatics.** We processed reads from our VESPA data sets with both QIIME 2[103] and DADA2 v.1.16.0[104] in the R environment v.3.6.3 and found that, while results were similar, DADA2 was more user-friendly (i.e., did not require installation of new software, required less steps, and was implementable within a familiar computing environment). Read files were converted to vectors and filtered for quality using the filterAndTrim command with default settings plus modifiers to remove primers (trimLeft = c(18,20)), residual PhiX reads (rm.phix = TRUE), and short sequences (minLen = 100). Error rate for forward and reverse reads were calculated using the learnErrors command, data were dereplicated using the derepFastq command, and Sequence Variants were inferred using the dada command. Read pairs were merged using the mergePairs command with justConcatenate = TRUE and chimeras were removed using the removeBimeraDenovo command with default parameters. Taxonomy assignments were made using the assignTaxonomy command and the PR$^2$ reference sequence database version 5.1.0 (https://doi.org/10.5281/zenodo.7805244), which contains 18S and 16S sequences at species-level resolution. For comparison we also tested two other taxonomy databases (DOI 10.5281/zenodo.1447329): v132 which includes all eukaryotic organisms from the SILVA v132 database and v128 which includes all eukaryotic organisms from the SILVA v128 database plus corrected species labels for *Blastocystis* and additional *Entamoeba* sequences. However, we found that the PR$^2$ database returned higher numbers of fully assigned ASVs. Any ASVs not assigned taxonomy using the PR$^2$ database were queried against the full NCBI nucleotide database on September 3rd, 2022 using MegaBLAST[105] with default parameters.

### Statistics and reproducibility

For PCR and metabarcoding experiments using purified DNA from single organisms or mock community standards, investigators were blinded to primer and template DNA identity, gels were scored by two independent investigators, and each experiment was successfully repeated with the same outcomes. For metabarcoding of clinical fecal samples, investigators were blinded to host species and infection status. Fecal amplicon PCR was performed successfully in triplicate and amplicons were pooled prior to library preparation. Sample sizes for fecal experiments were chosen opportunistically based on number of samples available from previous studies and no statistical method was used to predetermine sample size. No samples and no data were excluded from analysis of any experiment. Randomization was not used in this study because all samples were used for the same analysis, which was a comparison between methods.

For mock community standard analysis, statistical significance of mean fold distances from the theoretical input were derived from two-tailed Wilcoxon matched-pairs signed rank tests and Pearson correlation coefficients were calculated for the relationship between theoretical and observed read abundances. For clinical fecal sample analysis, richness and prevalence metrics were calculated for each sample and the statistical significance of differences in these metrics was determined using two-tailed Wilcoxon matched-pairs signed rank tests.

### Reporting summary

Further information on research design is available in the Nature Portfolio Reporting Summary linked to this article.

## Data availability

The DNA sequencing data generated in this study have been deposited in the National Center for Biotechnology Information (NCBI) Sequence Read Archive under BioProject ID PRJNA944233. The in silico PCR (% primer coverage) data, mock community standard analysis data, off-target read abundance data, and percent read abundance data generated in this study are provided in the Source Data file. Publicly available databases used in this study are: PR$^2$ reference sequence database v 5.0.1 (DOI: 10.5281/zenodo.7805244), the SILVA v128 and v132 dada2 formatted 18S 'train sets' (https://doi.org/10.5281/zenodo.1447329), and the SILVA SSU rRNA v138.1 Ref NR database (https://www.arb-silva.de/fileadmin/arb_web_db/release_138_1/ARB_files/SILVA_138.1_SSURef_

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

## Acknowledgements

We gratefully acknowledge Lyric Bartholomay, Timothy Yoshino, Peter Thompson, Monica Santin-Duran, Marie Pinkerton, Cecilia Escobar Lopez, Mostafa Zamanian, and Johanna Elfenbein for assistance in obtaining specimens, for cloning reagents, and for technical assistance. We thank Laura Parfrey for kindly providing the v132 and v128 taxonomy databases. This research was funded by National Institutes of Health (NIH) via grants 1R01AG049395-01 and 1R21AI163592-01 (T.L.G.) and the UW Madison Parasitology and Vector Biology Training Grant T32AI007414 (L.A.O.).

## Author contributions

T.L.G., S.F., L.A.O., conceived the study. T.L.G. and L.A.O. designed the study. L.A.O. designed the primers and collected and analyzed data. L.A.O. and T.L.G. wrote the manuscript. S.F. and B.M.D.G. provided data. S.F., B.M.D.G., L.J.K., M.C., O.N.-A., and M.G.D.-B. provided samples. All authors made substantive intellectual contributions, revised the manuscript, and approved the final draft.

## Competing interests

The authors declare no competing interests.
