## [Peer Review File · Nature Communications]

REVIEWER COMMENTS

Reviewer #1 (Remarks to the Author):

In this manuscript, the authors introduce what they describe as a new metabarcoding system for eukaryotes that they call VESPA, which is actually just a new primer pair for the 18S rRNA gene that performs well on a relatively small panel of known human pathogens (data analysis is done by existing methods such as DADA2, QIIME, and SILVA).

The technical accomplishment is worthwhile and the improvement in bias reduction over past primer pairs (for these specific taxa) is impressive. However, the paper should be clearer about what it is describing (a new primer pair and the PCR protocol that accommodates it, not a whole new metabarcoding system) and that it is specifically designed for taxa known to be human pathogens, not for all microbial eukaryotes. At least *in silico* performance should be described for a much broader range of microbial eukaryotes than is presented here, or the claims that it is broad-spectrum rather than pathogen-specific should be dropped.

The validation against synthetic communities and microscopy-examined specimens is a considerable strength of the paper and should be commended, although the number of sequences and samples is relatively hard. It should be recognized that this work is difficult to do and that ground-truthing against a large number of samples and against many synthetic communities would be challenging.

The taxonomic limitations of using 18S rRNA should be acknowledged more clearly (i.e. that it typically does not identify individual species), and at least some data on the ability of the primer pair to distinguish human parasites from related species that do not infect humans (or the limits of what can be distinguished) would be very useful. It would be especially valuable to see performance on an extended database to check how often the correct species is assigned, as taxonomic resolution is often a problem for 18S versus other markers that have been used for metabarcoding eukaryotes.

A minor comment is that rDNA does not exist (the ribosome does not contain DNA); "rRNA gene" is the correct nomenclature.

With a clearer focus on what is new in this paper vs commonly applied in the field, this will be a very useful addition to the literature for those seeking to assess parasites in human stool samples via 18S rRNA amplification.

Reviewer #2 (Remarks to the Author):

The topic of this manuscript is well defined and presented results are of high quality. The development and critical evaluation of the entire pipeline for performing characterization of eukaryotic endosymbionts is both timely and direly missing. The methodology is solid and well researched to yield the optimal version which is recommended, the languages and visualisation are clear and understandable.

I suggest addressing the following methodological issues:

- 1) EukMix is devised and proposed to become the community standard to test other protocols for quantification of eukaryotic endosymbionts, however this set contains equal amounts of taxa. The real-life scenario however, as demonstrated in Figs 3-4 shows, similar to microbiota, that different taxa often persist in more power-law distribution. Therefore ideally, a mock should be staggered as have previously been devised for the bacterial and viral communities (e.g. <https://doi.org/10.1128/mSystems.00062,10.1186/s40168-023-01533-x>). Please consider recreating a staggered mock and adding these results to the ones shown in Fig. 2.
- 2) The second methodological concern refers to the four primer sets that were removed due to off-target effects. Yet, in genomic DNA isolation, authors mention using standardised protocol that involved removing bacteria. Could this approach be applied to remove bacteria/archaeal DNA prior to amplification, increasing a list of primers to include these four sets, which I am sure would fail in later steps due to lower quality than the chosen primer set.

I see no other major issues, upon the points above being adequately addressed.

Reviewer #3 (Remarks to the Author):

The paper presents VESPA – a new a metabarcoding assays for surveying eukaryote endosymbionts. Based on a literature survey including 54 related papers, the authors chose the 4th variable region (V4) of the 18S marker gene undertook an in silico analysis of some of the most prominent previously published primer sets and sets of newly designed primers to assess the breadth of their phylogenetic

coverage. They also undertook experimental PCR testing of some of these primer sets using a panel of genomic DNA from 22 eukaryotic endosymbiont species. They go on to assess amplification biases with a set “mock community” samples made from cloned 18S DNA from 16 eukaryote endosymbiont species . They convincingly show that one of their primer sets (29F/21B8R) is a major improvement on any previous primer sets both in terms of eukaryotic endosymbiont coverage and lack of amplification bias. Finally they apply the VESPA protocol to a set of 12 human clinical samples and 40 non-human primate samples and compare the eukaryote endosymbiont species detected by VESPA compared to standard microscopy . They show VESPA is much more sensitive and captures a greater level of diversity in the samples. The work is essentially developing and validating a much improved “universal” primer set for eukaryotic endosymbionts (protozoa and helminths and microsporidia) .

In the paper’s favour; there is a need to progress in this area as this group of organisms is severely neglected in “microbiome” studies and there are still limited tools available. Consequently, this is an extremely useful study and I foresee these primers (and the information in the paper) as being very useful. Also the work appears well designed and performed overall , and is reasonably clearly presented throughout (except for my comments below). The data analysis seems solid and interpretation appropriate. It also represents a substantial amount of work.

The main weakness, for a high profile publication, is that testing of the assay on clinical samples is very limited (just 12 human sucrose floated samples) which doesn’t really seriously assess its value as a tool to look at the eukaryotic endosymbiont “biome” in different types of human sample. The biggest weakness of the VESPA assay is that its read depth is somewhat compromised when applied directly to stool DNA (due to fungi being efficiently targeted as explained more fully below) and this hasn’t really been investigated properly (or at all for human clinical samples). This is important because , if VESPA is to be used as it is being suggested as a standard high throughput universal eukaryotic endosymbiont assay (and/or in tandem with bacterial microbiome work), it would need to work well on human fecal stool DNA samples (rather than double centrifuged , sucrose floated organisms).

Major specific issues :

1. My biggest concern with the paper is the way the fungi are dealt with (or actually not dealt with!) . In the introduction, they say they “exclude” fungi as methods for that group have already been more extensively previously developed. However, if that’s the case, it is not clear to me why they then didn’t treat fungi as “off target” organisms like they did for bacteria and archaea in their primer design and in silico assessments. (It might be difficult or even impossible to avoid amplification of fungi with any universal eukaryotic microorganism primer set but it is not clear whether they assessed this at all?). Consequently, the 29F/21B8R primers actually seem to target fungi very well and so this means the application of the tool will be significantly compromised, in terms of read depth of target organisms, when applied directly to stool DNA due to the abundance of fungi in such samples. i.e. the majority of reads will be from fungi which is not what the VESPA assay is aimed at. This is well illustrated when the VESPA assay is applied to the 40 non-human primate fecal DNA samples – the majority of the reads are

from fungi (in some samples, almost all reads)(Figure 4A). (this was less apparent in the human samples as the DNA was made from eukaryotic endosymbionts purified from stool by sucrose flotation). Much more validation on different stool samples (particularly human) would be needed to assess the extent to which this is a limitation.

2. Why were the 12 clinical samples from humans sucrose floated organisms whereas the 40 non-human primates essentially stool DNA preps? Because of this , it is not clear how well the assay would function on human stool DNA samples (either in terms of PCR efficiency or in the fungal contamination issue described above). If human stool DNA samples are to be used for the VESPA assay in future then , based on the non-human primate data, (and the in silico specificity of the primers) , then there could be major problems of read depth due to fungal contamination . None of this was assessed or discussed. This seems a major omission given the stated value of the VESPA assay.

3. If sucrose floated organisms are to be used (like validated for the t human clinical samples here) instead of stool DNA samples, then there is a whole series of questions about whether all target organisms are appropriately captured (given that this is being presented as a “universal eukaryotic endosymbiont” assay). Also, it wouldn’t fit into simple workflows with 16S sequencing for bacteria and not be easily developed as a high throughput assay (still a useful assay but not in the way as presented in this paper as a high throughput universal assay)

4. It makes sense to target the V4 18S region – however, this will not allow phylogenetic resolution of many closely related eukaryotic species (why ITS regions are used for some groups). These limitations were not discussed.

=

Additional minor comments

5. Line 280: It is mentioned that their VESPA method helps resolve a cryptic species complex. It isn’t really easy to find this in the data – this could be explained a bit better

6. Line 316: why only use 18S and not “Short Sub-Unit (SSU)” which is very commonly used in the 18S literature? Might be missing some key papers

“VESPA: an optimized protocol for accurate metabarcoding-based characterization of vertebrate eukaryotic endosymbiont and parasite assemblages”

For publication in *Nature Communications*

Point-by-point responses to reviewer comments

Reviewer #1 (Remarks to the Author):

Comment 1: *In this manuscript, the authors introduce what they describe as a new metabarcoding system for eukaryotes that they call VESPA, which is actually just a new primer pair for the 18S rRNA gene that performs well on a relatively small panel of known human pathogens (data analysis is done by existing methods such as DADA2, QIIME, and SILVA).*

Response 1: We thank the reviewer for raising this important point. We agree that our use of “pipeline” could be misleading because it may imply that all aspects of VESPA metabarcoding are novel, which is not the case. As stated in comment 2 by the reviewer, one novel aspect is indeed the primers. We also extensively tested each step of the metabarcoding protocol (DNA extraction method, PCR polymerases, PCR parameters, library preparation methods, data analysis tools) and therefore believe it is appropriate to claim that we are describing an optimized metabarcoding protocol for use with the primers. To address this point, we have revised the manuscript to clearly state which aspects are entirely novel (primers) and which are optimized (metabarcoding protocol). To this end, we replaced “an optimized pipeline” in the title with “an optimized protocol” (lines 1 – 3) and changed all instances of “pipeline” in the body of the manuscript to “primers and optimized protocol” (lines 35 – 37, 93 – 94). We also changed all instances of “method” when referring specifically to VESPA, to “protocol” or “primers and optimized protocol” for further clarity (lines 98, 107, 114, 257, 268, 275, 283, 311, 337). We also agree that our protocol does not detect all eukaryotes. Please see our response to comment 2 below where we address this issue.

Comment 2: *The technical accomplishment is worthwhile and the improvement in bias reduction over past primer pairs (for these specific taxa) is impressive. However, the paper should be clearer about what it is describing (a new primer pair and the PCR protocol that accommodates it, not a whole new metabarcoding system) and that it is specifically designed for taxa known to be human pathogens, not for all microbial eukaryotes. At least in silico performance should be described for a much broader range of microbial eukaryotes than is presented here, or the claims that it is broad-spectrum rather than pathogen-specific should be dropped.*

Response 2: Many thanks for the positive appraisal and for bringing up these valid points. As stated above, we agree that the language should be clearer to delineate the novel aspect (primers) of the work. Please see our response to comment 1 for wording changes to address this issue. We also agree that our protocol is not designed for use with all eukaryotes or all microbial eukaryotes. We designed and tested the primers and protocol

with vertebrate eukaryotic endosymbionts (thus, the VESPA acronym), which we define as including macro- and microscopic, pathogenic and non-pathogenic organisms residing within animal and human hosts. We purposely did not include free-living microbial eukaryotes in our primer design in order to gain better coverage of host-associated target groups. In order to address this critique, we clarified the wording of the manuscript. We changed “eukaryotic microbiota” to “host-associated eukaryotes” (lines 32 – 33), corrected the use of “eukaryotic assemblages” to “host-associated eukaryotic assemblages” (line 39), and modified “helminths and protozoa” to “host-associated helminths and protozoa” (lines 171 – 172, 906).

Comment 3: *The validation against synthetic communities and microscopy-examined specimens is a considerable strength of the paper and should be commended, although the number of sequences and samples is relatively hard. It should be recognized that this work is difficult to do and that ground-truthing against a large number of samples and against many synthetic communities would be challenging.*

Response 3: We thank the reviewer for the encouraging comment. In response to other reviewers’ comments, we increased the number of human samples by 40 and added analysis of a second mock community.

Comment 4: *The taxonomic limitations of using 18S rRNA should be acknowledged more clearly (i.e. that it typically does not identify individual species), and at least some data on the ability of the primer pair to distinguish human parasites from related species that do not infect humans (or the limits of what can be distinguished) would be very useful. It would be especially valuable to see performance on an extended database to check how often the correct species is assigned, as taxonomic resolution is often a problem for 18S versus other markers that have been used for metabarcoding eukaryotes.*

Response 4: This is an excellent point. We acknowledge that there are well-documented taxonomic limitations to using 18S. Our new primers target the same region of 18S (V4-V5) as several other published metabarcoding primer sets, including all of those that we tested in the study. We provide evidence that the new primers have better coverage than existing primers due to their specific sequences, but since the amplified region is the same, the taxonomic resolution is the same as published primer sets. Because we are not addressing choice of marker, we did not include comparisons of other markers as this was beyond the scope of the study. To clarify this issue, we added a section to the Discussion to this effect with additional references to further justify our choice of 18S V4 for amplification (lines 289 – 293).

Comment 5: *A minor comment is that rDNA does not exist (the ribosome does not contain DNA); “rRNA gene” is the correct nomenclature.*

Response 5: Good catch! We have corrected rDNA to rRNA (line 437 and Figure 2a).

Comment 6: *With a clearer focus on what is new in this paper vs commonly applied in the field, this will be a very useful addition to the literature for those seeking to assess parasites in human stool samples via 18S rRNA amplification.*

Response 6: We thank the reviewer for the careful consideration of our manuscript and thoughtful critiques. We agree that our manuscript would benefit from a clearer focus. Please see responses to comments 1 - 4 above for how we addressed this and other related issues.

Reviewer #2 (Remarks to the Author):

Overall comment: *The topic of this manuscript is well defined and presented results are of high quality. The development and critical evaluation of the entire pipeline for performing characterization of eukaryotic endosymbionts is both timely and direly missing. The methodology is solid and well researched to yield the optimal version which is recommended, the languages and visualisation are clear and understandable.*

Response: Many thanks for the positive comments.

Comment 1: *I suggest addressing the following methodological issues: EukMix is devised and proposed to become the community standard to test other protocols for quantification of eukaryotic endosymbionts, however this set contains equal amounts of taxa. The real-life scenario however, as demonstrated in Figs 3-4 shows, similar to microbiota, that different taxa often persist in more power-law distribution. Therefore ideally, a mock should be staggered as have previously been devised for the bacterial and viral communities (e.g. [https://doi.org/10.1128/mSystems.00062, 10.1186/s40168-023-01533-x](https://doi.org/10.1128/mSystems.00062.10.1186/s40168-023-01533-x)). Please consider recreating a staggered mock and adding these results to the ones shown in Fig. 2.*

Response 1: The reviewer makes an excellent point. We have conducted the recommended experiments. Specifically, we created a staggered mock community (following a logarithmic function based on theoretical and empirical research and with abundances spanning 400-fold) with the same 16 organisms as the equimolar mock community and performed metabarcoding with the same 5 primers as in the manuscript. We have added the results of the analysis to Figure 2 as the reviewer suggested (lines 961 – 973) along with explanatory text in the Results (lines 174 – 189), Discussion (lines 278 – 282), and Methods (lines 447 – 456, 458, 464 – 466) sections.

Comment 2: *The second methodological concern refers to the four primer sets that were removed due to off-target effects. Yet, in genomic DNA isolation, authors mention using standardised protocol that involved removing bacteria. Could this approach be applied to remove bacteria/archaeal DNA prior to amplification, increasing a list of primers to include these four sets, which I am sure would fail in later steps due to lower quality than the chosen primer set.*

Response 2: We are grateful to the reviewer for this very good suggestion. The reviewer is correct that standardized methods generally remove/reduce bacteria. In fact, there are a great many such methods that have been described in the literature. It would indeed be

possible to apply such methods to samples prior to implementing our method. However, due to the sheer number of methods that have been used, it would increase the complexity of this experiment by tens or hundreds of folds. We have therefore added a statement to the Methods section stating that methods to reduce bacteria and archaea would likely work in concert with our method (lines 503 – 509).

Concluding comment: *I see no other major issues, upon the points above being adequately addressed.*

Response: We thank the reviewer for the helpful critiques and suggestions.

Reviewer #3 (Remarks to the Author):

Comment 1: *The paper presents VESPA – a new metabarcoding assays for surveying eukaryote endosymbionts. Based on a literature survey including 54 related papers, the authors chose the 4th variable region (V4) of the 18S marker gene undertook an in silico analysis of some of the most prominent previously published primer sets and sets of newly designed primers to assess the breadth of their phylogenetic coverage. The also undertook experimental PCR testing of some of these primers sets using a panel of genomic DNA from 22 eukaryotic endosymbiont species. They go onto assess amplification biases with a set “mock community” samples made from of cloned 18S DNA from 16 eukaryote endosymbiont species . They convincingly show that one of their primer sets (29F/21B8R) is a major improvement on any previous primers sets both in terms of eukaryotic endosymbiont coverage and lack of amplification bias. Finally they apply the VESPA protocol to a set of 12 human clinical samples and 40 non-human primate samples and compare the eukaryote endosymbiont species detected by VESPA compared to standard microscopy . They show VESPA is much more sensitive and captures a greater level of diversity in the samples. The work is essentially developing and validating a much improved “universal” primer set for eukaryotic endosymbionts (protozoa and helminths and microsporidia) .*

In the paper’s favour; there is a need to progress in this area as this group of organisms is severely neglected in “microbiome” studies and there are still limited tools available. Consequently, this is an extremely useful study and I foresee these primers (and the information in the paper) as being very useful. Also the work appears well designed and performed overall , and is reasonably clearly presented throughout (except for my comments below). The data analysis seems solid and interpretation appropriate. It also represents a substantial amount of work.

Response 1: We thank the reviewer for the thoughtful and positive appraisal of our work.

Comment 3: *The main weakness, for a high profile publication, is that testing of the assay on clinical samples is very limited (just 12 human sucrose floated samples) which doesn’t really seriously assess its value as a tool to look at the eukaryotic endosymbiont “biome” in different types of human sample. The biggest weakness of the VESPA assay is that its read depth is somewhat comprised when applied directly to stool DNA (due to fungi being efficiently targeted as explained more fully below) and this hasn’t really been investigated properly (or at all for human clinical samples). This is important because , if VESPA is to be used as it is being suggested as a standard a high throughput universal eukaryotic endosymbiont assay (and/or in tandem with*

bacterial microbiome work), it would need to work well on human fecal stool DNA samples (rather than double centrifuged , sucrose floated organisms).

Response 3: The reviewer makes an excellent point and is clearly knowledgeable about high throughput metabarcoding assays. We agree that VESPA must perform well on whole fecal DNA, not just sucrose floated samples, in order to be of broad use. We further acknowledge our lack of explanation of this incongruity in the manuscript. For validating VESPA, we chose these samples because they were collected from a remote population with an extraordinary diversity of eukaryotic endosymbionts and had already been thoroughly characterized by expert parasitologists, which made them unique and valuable for testing our new primers and protocol. The samples were collected as part of a previously conducted study and were not ideal in that they had to be processed to remove bacteria prior to shipping, but we did not have access to whole fecal samples from this population. To clarify this point we have added additional explanation and justification for use of these samples to the Methods section (lines 503 – 509).

Major specific issues:

Comment 1: *My biggest concern with the paper is the way the fungi are dealt with (or actually not dealt with!). In the introduction, they say they “exclude” fungi as methods for that group have already been more extensively previously developed. However, if that’s the case, it is not clear to me why they then didn’t treat fungi as “off target” organisms like they did for bacteria and archaea in their primer design and in silico assessments. (It might be difficult or even impossible to avoid amplification of fungi with any universal eukaryotic microorganism primer set but it is not clear whether they assessed this at all?. Consequently, the 29F/21B8R primers actually seem to target fungi very well and so this means the application of the tool will be significantly compromised, in terms of read depth of target organisms, when applied directly to stool DNA due to the abundance in fungi in such samples. i.e. the majority of reads will be from fungi which is not what the VESPA assay is aimed at. This is well illustrated when the VESPA assay is applied to the 40 non-human primate fecal DNA samples – the majority of the reads are from fungi (in some samples, almost all reads)(Figure 4A). (this was less apparent in the human samples as the DNA was made from eukaryotic endosymbionts purified from stool by sucrose flotation). Much more validation on different stool samples (particularly human) would be needed to assess the extent to which this is a limitation.*

Response 1: Many thanks for the thoughtful critique. In light of the reviewer’s excellent suggestions, we have performed additional experiments. We examined the *in silico* coverage of fungi across primer sets and found that all 18S V4 primers, new and published, had high coverage of fungal groups. This was not unexpected based on 18S sequence similarity between eukaryotic endosymbionts and fungi and because all of the primers in question broadly target eukaryotes. We have added these data as an additional column in Table 2 (lines 910 – 916) and have added corresponding text in the Results section (lines 151 – 153). To address the issue of off-target amplification and read depth, we applied VESPA metabarcoding to 40 whole, unprocessed human fecal samples and quantified levels of off-target read abundance, including fungal reads. Overall there was low abundance of prokaryotic, host, and fungal reads (total < 50 % of reads in every case)

despite starting with whole fecal samples. We have added a figure (Figure 3, lines 974 – 980) displaying these results for each sample and as an overall mean +/- SEM and added corresponding text to the Abstract (lines 39 – 40), Introduction (line 97), Results (lines 110, 192 – 200) and Discussion (lines 265 – 268) sections. The overall levels of fungal reads varied in human samples (lower) versus non-human primate samples (higher) which we believe may be due to differences in fecal composition between hosts. We have added a section to the Discussion to this effect including relevant references (lines 294 – 302).

Comment 2: *Why were the 12 clinical samples from humans sucrose floated organisms whereas the 40 non-human primates essentially stool DNA preps? Because of this, it is not clear how well the assay would function on human stool DNA samples (either in terms of PCR efficiency or in the fungal contamination issue described above). If human stool DNA samples are to be used for the VESPA assay in future then, based on the non-human primate data, (and the in silico specificity of the primers), then there could be major problems of read depth due to fungal contamination. None of this was assessed or discussed. This seems a major omission given the stated value of the VESPA assay.*

Response 2: We thank the reviewer for raising this important point. The 12 human clinical samples were from sucrose floated organisms because, unfortunately, this was the only sample type available to us from this population, which met our study criteria (see above). Although we suggest using the VESPA protocol on whole, unprocessed fecal samples, we used these processed (floated) samples for the above reasons, but, as the reviewer points out, we omitted this methodological caveat in the original manuscript. To rectify this issue, we have added the missing details and caveats to the Methods section (lines 503 – 509).

Comment 3: *If sucrose floated organisms are to be used (like validated for the t human clinical samples here) instead of stool DNA samples, then there is a whole series of questions about whether all target organisms are appropriately captured (given that this is being presented as a “universal eukaryotic endosymbiont” assay). Also, it wouldn’t fit into simple workflows with 16S sequencing for bacteria and not be easily developed as a high throughput assay (still a useful assay but not in the way as presented in this paper as a high throughput universal assay)*

Response 3: We are grateful to the reviewer for making this critical point. We do not suggest using sucrose floated organisms for exactly the reasons delineated by the reviewer here and have added a statement to the Methods section to this effect (lines 503 – 505).

Comment 4: *It makes sense to target the V4 18S region – however, this will not allow phylogenetic resolution of many closely related eukaryotic species (why ITS regions are used for some groups). These limitations were not discussed.*

Response 4: This is an excellent point which was also brought up by reviewer 1 above. We agree that using 18 V4 will not taxonomically resolve all eukaryotic endosymbionts to the species level and that this limitation needs to be included in the manuscript. We have added a section to the discussion to address this issue (lines 289 – 293).

Additional minor comments:

Comment 5: Line 280: It is mentioned that their VESPA method helps resolve a cryptic species complex. It isn't really easy to find this in the data – this could be explained a bit better

Response 5: Thanks to the reviewer for drawing our attention to this point. We agree that this aspect of the text would benefit from a clearer explanation. To clarify the resolution of the *Entamoeba histolytica/dispar* cryptic species complex, we have added explanatory text to the Results section (line 242) and Discussion section (lines 304 – 305) and have highlighted the associated data with asterisks in Figure 5 (lines 1027 – 1029).

Comment 6: Line 316: why only use 18S and not “Short Sub-Unit (SSU)” which is very commonly used in the 18S literature? Might be missing some key papers.

Response 6: Great point! Based on the reviewer's excellent suggestion, we have performed an updated literature review including “SSU,” “short sub-unit” and variants thereof in the search terms, but this effort recovered only the papers previously identified. We now include the additional suggested search terms in the Methods section (lines 341 – 342).

REVIEWER COMMENTS

Reviewer #1 (Remarks to the Author):

This revised version of the manuscript is substantially improved, especially with the addition of the analysis of complete stool samples.

However, it is essential that the authors better address the taxonomic resolution that they are able to achieve in different groups of eukaryotes, rather than just saying that it won't always be at the species level, as often the groups that are identical over this region will be quite large, especially after protocols such as DADA2 are applied to reduce noise. This really needs to be analyzed carefully and the results clearly signaled to readers in the main text for the work to be appropriate for NCOMMS.

Reviewer #3 (Remarks to the Author):

I have reviewed the author responses and the revised manuscript and I am satisfied they have addressed my comments

“VESPA: an optimized protocol for accurate metabarcoding-based characterization of vertebrate eukaryotic endosymbiont and parasite assemblages”

For publication in *Nature Communications*

Point-by-point responses to reviewer comments

Reviewer #1 (Remarks to the Author):

Comment 1: This revised version of the manuscript is substantially improved, especially with the addition of the analysis of complete stool samples.

Response 1: We thank the reviewer for the encouraging comments.

Comment 2: However, it is essential that the authors better address the taxonomic resolution that they are able to achieve in different groups of eukaryotes, rather than just saying that it won't always be at the species level, as often the groups that are identical over this region will be quite large, especially after protocols such as DADA2 are applied to reduce noise. This really needs to be analyzed carefully and the results clearly signaled to readers in the main text for the work to be appropriate for NCOMMS.

Response 2: Many thanks for the thoughtful critique. In light of the reviewer's excellent suggestions, we have performed additional analyses to assess the issue of taxonomic resolution. We carried out *in silico* PCR with our VESPA primers against a parasite 18S database, assigned taxonomy with the IDs from the same database, and quantified the number of amplicons resolved to the species or genus level. The resolution was better in protozoal groups compared to helminth groups, but overall resolution was high, with species-level identification rates ranging from 94.6 to 99.7 %. The resulting data have been added as a table (Table 3, lines 968 - 970) with explanatory text added to the Results (lines 160 - 166), Discussion (lines 297 - 306) and Methods (lines 430 - 441) sections. We have also included a comprehensive table of unresolved sequences in the supplement (Supplementary Table 2).

Reviewer #2 (Remarks to the Author):

Overall comment: I have reviewed the author responses and the revised manuscript and I am satisfied they have addressed my comments

Response: We are grateful to the reviewer for the positive appraisal.